# Connexin Genes Variants Associated with Non-Syndromic Hearing Impairment: A Systematic Review of the Global Burden

**DOI:** 10.3390/life10110258

**Published:** 2020-10-28

**Authors:** Samuel Mawuli Adadey, Edmond Wonkam-Tingang, Elvis Twumasi Aboagye, Daniel Wonder Nayo-Gyan, Maame Boatemaa Ansong, Osbourne Quaye, Gordon A. Awandare, Ambroise Wonkam

**Affiliations:** 1West African Centre for Cell Biology of Infectious Pathogens (WACCBIP), University of Ghana, P.O. Box LG 54, Legon GA184, Accra, Greater Accra Region, Ghana; smadadey@st.ug.edu.gh (S.M.A.); oquaye@ug.edu.gh (O.Q.); gawandare@ug.edu.gh (G.A.A.); 2Department of Biochemistry, Cell and Molecular Biology, University of Ghana, P.O. Box LG 54, Legon Accra GA184, Greater Accra Region, Ghana; atelvis45@gmail.com (E.T.A.); boatiesong@gmail.com (M.B.A.); 3Division of Human Genetics, Faculty of Health Sciences, University of Cape Town, Anzio Road, Observatory, Cape Town 7925, South Africa; wonkamedmond@yahoo.fr; 4Department of Applied Chemistry and Biochemistry, C. K. Tedam University of Technology and Applied Sciences, P.O. Box 24, Navrongo 00000, Upper East Region, Ghana; dwondernayo@gmail.com

**Keywords:** connexin, gap junction protein, gene variant, *GJB2*, systematic review

## Abstract

Mutations in connexins are the most common causes of hearing impairment (HI) in many populations. Our aim was to review the global burden of pathogenic and likely pathogenic (PLP) variants in connexin genes associated with HI. We conducted a systematic review of the literature based on targeted inclusion/exclusion criteria of publications from 1997 to 2020. The databases used were PubMed, Scopus, Africa-Wide Information, and Web of Science. The protocol was registered on PROSPERO, the International Prospective Register of Systematic Reviews, with the registration number “CRD42020169697”. The data extracted were analyzed using Microsoft Excel and SPSS version 25 (IBM, Armonk, New York, United States). A total of 571 independent studies were retrieved and considered for data extraction with the majority of studies (47.8% (*n* = 289)) done in Asia. Targeted sequencing was found to be the most common technique used in investigating connexin gene mutations. We identified seven connexin genes that were associated with HI, and *GJB2* (520/571 publications) was the most studied among the seven. Excluding PLP in *GJB2*, *GJB6*, and *GJA1* the other connexin gene variants (thus *GJB3*, *GJB4*, *GJC3*, and *GJC1* variants) had conflicting association with HI. Biallelic *GJB2* PLP variants were the most common and widespread variants associated with non-syndromic hearing impairment (NSHI) in different global populations but absent in most African populations. The most common *GJB2* alleles found to be predominant in specific populations include; p.Gly12ValfsTer2 in Europeans, North Africans, Brazilians, and Americans; p.V37I and p.L79Cfs in Asians; p.W24X in Indians; p.L56Rfs in Americans; and the founder mutation p.R143W in Africans from Ghana, or with putative Ghanaian ancestry. The present review suggests that only *GJB2* and *GJB3* are recognized and validated HI genes. The findings call for an extensive investigation of the other connexin genes in many populations to elucidate their contributions to HI, in order to improve gene-disease pair curations, globally.

## 1. Introduction

Hearing impairment (HI) is the most common sensorineural disability worldwide, with a global prevalence of 1.3 per 1000 population [1,2]. It occurs in about 1 per 1000 live births in high-income countries, with a much higher incidence of up to 6 per 1000 in the lower-income countries [3]. According to the World Health Organization, 466 million people are living with HI and about 900 people will be affected by the year 2050 [4]. Depending on the degree of severity, HI can be classified as mild, moderate, severe, or profound when the pure tone average ranges from 26 to 40 dB, 41 to 60 dB, 61 to 80 dB or is over 81 dB, respectively (Deafness and Hearing Loss, n.d.). It is estimated that approximately 50% of congenital profound HI cases are of genetic origin [5]. If there are no other distinguishing clinical findings, HI is classified as non-syndromic [6]. About 80% of non-syndromic HI (NSHI) cases are inherited in an autosomal recessive mode, while an autosomal dominant pattern of inheritance is observed in 18% of cases [7]. In the remaining 2% of cases, the mode of inheritance is either X-linked or mitochondrial [7].

Non-syndromic HI is extremely heterogeneous, with approximately 170 loci and 121 genes identified so far [8]. Studies in European and Asian populations have identified mutations in connexin genes as the major contributors to NSHI [9,10]. Connexins (Cx) are a homogeneous family of proteins expressed in a large variety of tissues in the human body and known for their assembly into intercellular channels, called gap junctions [11]. Twenty-one different human connexin genes have been reported so far, each coding for a transmembrane protein with the same protein topology [11]. Connexins have four transmembrane domains (TM), TM1–TM4, connected by two extracellular loops (E), E1, and E2, which mediate docking [12]. The N- and C-termini, and a loop connecting TM2 and TM3 are on the cytoplasmic side of the plasma membrane [12].

Connexins are synthesized in the endoplasmic reticulum (ER) and oligomerize in the ER/Golgi or trans-Golgi network to form hexameric hemichannels or connexons [11]. Connexons are transported to the plasma membrane, where they can act as functional channels by themselves, or move to regions of cell contact and find a partner hemichannel from an adjacent cell to form a complete gap junction channel [12]. Gap junctions play an important role in cell-cell communication and homeostasis in various tissues, by mediating a direct exchange of ions and other small molecules up to 1 kDa (including a variety of second messengers, metabolites, but also small linear peptides) between the cytoplasms of adjacent cells [11].

To date, mutations in four connexin genes including *GJB2* (Cx26), *GJB3* (Cx31), *GJB4* (Cx30.3), and *GJB6* (Cx30) have been associated with sensorineural HI [13,14,15]. These four connexins were shown to be expressed in the inner ear, and some studies supported their role in potassium removal and recycling in the ear, as well as a possible role for nutrient passage in the cochlea [16]. *GJB2*-related sensorineural HI can occur alone or in association with hyperproliferative skin disorders, as in the case in Keratitis-ichthyosis-deafness syndrome and Bart-Pumphrey syndrome [17,18,19]. It is has been shown that digenic inheritance of recessive deafness by mutations in *GJB2* and *GJB6*, or *GJB2* and *GJB3* can occur [11]. In other words, deafness can be caused by the addition of a mutation in one allele of GJB2 and one allele of *GJB6* or *GJB3*, indicating an interaction of these connexins in the cochlea [11]. *GJB6* coding region variants have been proven not to cause HI using mouse models, however, the large deletions of the *GJB6* gene especially *GJB6*-D13S1830 were implicated as causal factors of HI. The cis-acting element upstream of *GJB2* and *GJB6* gene is disrupted by the large genomic deletions abolishing the expression of *GJB2* gene which is responsible for the development of HI [20].

*GJB2* and *GJB6* genes have been well studied in Europeans and Asians, with c.35delG identified as the most prevalent *GJB2* mutations associated with NSHI [9]. However, the other NSHI-causing connexin genes (i.e., *GJB3* and *GJB4*) have not been extensively studied [11,21]. Using a systematic review approach, we provided summary data on connexin gene variants associated with HI, and specifically the global contribution of connexin genes to NSHI.

## 2. Results

Of the 2592 studies that were screened, 571 articles were downloaded and analyzed. The 571 articles comprised publications that dated as far back as 1997 to recent publications in 2020 (Figure 1A). The analysis suggested that there were few studies on connexin gene variants association with HI in first the three years of the study timeframe (1997, 1998, and 1999), followed by a drastic increase in the last two decades. The year 2015 recorded the highest number of publications on connexin gene variants (Figure 1A).

Most of the articles retrieved were from Asia (47.8% (*n* = 289)) with China (99/289) recording the highest number of articles. Australia had the least number (0.7% (*n* = 4)) of retrieved articles (Figure 1B,C). There were relatively few studies from Africa (6% (*n* = 35), compared to other continents (Figure 1). Asia had reports on all connexins (7) found in this review, while Europe and Australia reported on 4 connexins. Africa, North America, and South America had reports of 6, 5, and 3 connexins respectively (Figure 1D). We identified *GJB2* as the most widely studied connexin in all the continents (Figure 1D).

We identified a cocktail of methods used by the researchers to investigate connexin gene variants in hearing-impaired patient samples. Several studies employed two or more approaches while others depended on a single approach. Targeted sequencing was the most common method followed by polymerase chain reaction (PCR) techniques. The targeted sequencing studies were mostly by Sanger sequencing where one or more primer sets were used to amplify and sequence the coding regions of the connexins under investigation. The analysis of the primer sets used for sequencing revealed their uniqueness for each study (Appendix A). Among the PCR techniques were amplification-refractory mutation system (ARMS), PACE™ (PCR Allele Competitive Extension), and multiplex PCR. A few studies used next-generation sequencing techniques (NGS) such as whole-exome sequencing (NGS) and NGS panel sequencing (Figure 2).

### 2.1. Connexin 26 (GJB2)

We identified 337 variants in *GJB2* from the review of the publications included in this study. Analysis of the clinical significance of these variants gave 124 (37.2%), 53 (15.9%), 46 (13.8%), and 25 (7.5%) pathogenic, likely pathogenic, uncertain significance, and benign variants, respectively (Figure 3). Asia (*n* = 288; 48%) was the highest contributor of pathogenic (PLP) variants followed by Europe (*n* = 166; 28%), North America (*n* = 52; 9%), Africa (*n* = 48, 8%); South America (*n* = 37; 6%) and Australia (*n* = 4; 1%). The three databases used in this study did not have data on the clinical significance of 13 out of the 333 variants, implying that some of these variants may be novel (Appendix A).

The pathogenic variants were sorted based on the number of reported alleles to identify the commonly reported variants. The variants with more than 450 reported alleles were considered as the commonly associated *GJB2* variants. Based on the number of reported alleles, p.Gly12ValfsTer2 (c.35delG) was ranked as the most commonly reported *GJB2* variant, found in 26,429 (15.1%) out of 175,491 investigated alleles. The frequencies of the other common *GJB2* mutations were 10,009/82,805 (12.1%), 9813/277,116 (3.5%), 3520/127,802 (2.8%), 972/54,394 (1.8%), 641/54,279 (1.2%), 1080/10,7855 (1.0%), and 497/90,305 (0.6%) for p.M34T (c.101T > C), p.L79Cfs (c.235delC), p.V37I (c.109G > A), p.W24X (c.71G > A), p.L56Rfs (c.167delT), p.H100RfsTer14 (c.299_300delAT), and p.R143W (c.427C > T), respectively (Appendix A). Further analysis of the p.Gly12ValfsTer2 (c.35delG) showed its widespread and high prevalence in European countries, and North African countries and some parts of Brazil and America. In Asia, p.V37I (c.109G > A and p.L79Cfs (c.235delC)) variants were the most frequently reported *GJB2* variant. On a country-wise analysis, we observed that the highest number of p.Gly12ValfsTer2 (c.35delG) and p.M34T (c.101T > C) mutated alleles were from the United Kingdom (Figure 4). The highest number of mutated alleles for p.L79Cfs (c.235delC), p.V37I (c.109G > A), and p.H100RfsTer14 (c.299_300delAT) were recorded in China. India, the United States of America, and Ghana recorded the highest number of p.W24X (c.71G > A), p.L56Rfs (c.167delT), and p.R143W (c.427C > T) mutated alleles, respectively (Figure 4).

We extracted data on known PLP variants in *GJB2* (p.W44*: c.131G > A (North America), c.IVS1 + 1G > A (Russia), p.W172*: c.516G > A (Siberia), p.W172C: c.516G > C (Siberia) p.W172R, and c.514T>A (Siberia) that are prevalent in isolated ethnic groups. The majority of mutated c.IVS1 + 1G > A variant was recorded in Asia with high frequencies from Yakutia, Siberia, and Russia (Appendix A). The other three rare *GJB2* variants (p.W172*: c.516G > A, p.W172C: c.516G > C, p.W172R, and c.514T > A) which are at the same amino acid position were common in the Asian countries with Rusia having the highest frequency (Appendix A).

### 2.2. Connexin 30 Gene (GJB6)

We selected reports of 18 variants in *GJB6*, including two large genomic deletions. Most coding region variants in *GJB6* were predicted as benign or uncertain significance (*n* = 12 (85.7%)). Two variants were predicted as pathogenic; *GJB6*: p.A40V/c.119C > T/rs780320724 from Taiwan [23] and *GJB6*: p.T5M/c.14C > T/rs104894414 from Germany [24], and Iran [25], respectively (Table 1). The del(*GJB6*-D13S1830) was the most common *GJB6* variant reported from 31 countries from all continents, with virtually no case from Africa. High allele frequencies were particularly reported in France and Spain (Figure 5). We found few studies that reported del(*GJB6*-D13S1854) variation among the hearing-impaired; these studies were from Argentina, Colombia, Portugal, and Brazil.

### 2.3. Connexin 31 Gene (GJB3)

We identified reports of variants of *GJB3* from nine countries with Korea having the highest number of reported alleles. The variant with the highest number of reported alleles was found to be synonymous and was predicted to be benign. None of the variants was predicted pathogenic although they were identified in hearing-impaired populations. Based on the databases used, we predicted the variants as benign, with uncertain significance or conflicting interpretations. Two variants (c.547G > A/rs74315318 and c.497A > G/rs121908851) were predicted as pathogenic by only one database with conflicting interpretations from the other databases; it was, therefore, difficult to conclude on their pathogenicity (Table 2).

### 2.4. Connexin 30.3 Gene (GJB4)

In this review, we identified studies from 5 countries that reported *GJB4* variants. The majority of identified *GJB4* variants had conflicting clinical significance since they had different interpretations in the databases used. VarSome predicted seven variants as pathogenic or likely pathogenic and InterVar predicted an additional variant as pathogenic. The pathogenic variants of VarSome were not predicted as pathogenic by InterVar and vice versa (Table 3).

### 2.5. Connexin 29 Gene (GJC3)

We identified seven *GJC3* variants from 4 countries (Table 4) which were predicted as benign or of uncertain significance with no PLP variant found.

### 2.6. Connexin 43 Gene (GJA1)

Twelve (12) variants in *GJA1* were reported by different researchers of which three were predicted to be pathogenic or likely pathogenic (PLP). The pathogenic variants were reported in hearing-impaired patients from America and Asia, the variants reported from Africa (South Africa) were predicted as benign or uncertain significance. *GJA1* c.932delC variant had the highest number of alleles reported from Australia. Four (4) out of the 11 *GJA1* variants were reported in a study from South Africa (Table 5).

### 2.7. Connexin 45 (GJC1)

Connexin 45 was previously given the gene symbol *GJA7* but the Human Genome Organization (HUGO) Gene Nomenclature Committee [51] has given it the symbol *GJC1.* We identified a multi-site study that reported *GJC1* variants from three different populations: the USA, Turkey, and the UK [52]. None of the reported variants were predicted pathogenic although they were identified in hearing-impaired cohorts. We observed conflicting interpretations of the variants’ clinical significance based on the predictions from the databases used (Table 6).

### 2.8. Summary of the Global Allele Frequencies of the Common Connexin Genes Pathogenic (PLP) Variants Associated to Hearing Impairment (HI)

We calculated the allele frequency of the reported PLP variants in connexin genes in patients and controls for each continent, in order to assess the global contribution of these variants associated with HI. Asia had the highest allele frequencies of the common *GJB2* variants in the hearing controls compared to the other continents. *GJB2:*p.V37I:c.109G > A had the highest allele frequency (200/3478 (5.8%)) among the control group from Asia (200/3478 (5.8%)) and North America (11/588 (1.9%)). In Africa, *GJB2*:p.Gly12ValfsTer2:c.35delG and p.R143W:c.427C > T recorded the highest frequencies in Ghana. In Europe, among the eight common *GJB2* variants, p.Gly12ValfsTer2:c.35delG (Table 7). We did not find reports of carriers of the *GJB6* large deletions and *GJA1* PLP variants.

## 3. Discussion

To the best of our knowledge, this paper is, to date, the most comprehensive review on the contribution of connexin gene variants in HI, globally. Connexin channels regulate the transport of small signaling molecules between cells to aid the proper functioning of the tissue/organ systems in the body [53]. Our review found that more than 570 studies were conducted globally on connexin-related HI investigations with most studies performed in Asia, while relatively few have been done in Africa.

Most studies used targeted sequencing, but the decline in next-generation sequencing cost has accelerated the discovery of novel disease gene variants through available high-throughput targeted panels or whole-exome sequencing technologies investigating several gene targets in a single test [54,55,56]. Indeed, there was a clear migration from non-sequencing approaches such as denaturing high-performance liquid chromatography (DHPLC), multiplex ligation-dependent probe amplification (MLPA), PCR, restriction fragment length polymorphism (RFLP), and single-strand conformational polymorphism (SSCP) to sequencing techniques or a combination of sequencing and non-sequencing techniques.

Connexin 26 gene (*GJB2*, OMIM:121011) located on chromosome 13q12.11 is known to be expressed in different tissues including the cochlear of humans [57], mouse, and rat [58]. *GJB2* gene variants were the most common genetic factors associated with NSHI among several populations [59,60], however, they are rare in African, and African American populations [61]. Similarly, it was clear from our review that *GJB2* is the most investigated gene and had the highest number of pathogenic variants identified among hearing-impaired patients. The most common pathogenic variants (*GJB2*: p.Gly12ValfsTer2, p.M34T, p.L79Cfs, p.V37I, p.H100RfsTer14, p.W24X, p.L56Rfs, and p.R143W) appeared to be localized to specific populations, due to a founder effect [10,62].

In a previous review by Chan and Chang in 2014, 216 original *GJB2* research articles reporting not less than 10 probands were retrieved and analyzed [10]. In our current review, 571 original research publications on connexins associated with HI were considered of which 566 articles reported on *GJB2* associated HI. The previous report was from 63 countries [10], while in this study, we retrieved *GJB2* publications from 106 countries. The differences in the number of publications and countries involved can be explained by the time difference between the previous report and the present study. Also, we did not exclude case reports, contrary to the previous report. In contrast to the report from Chan and Chang, Australia, and not Africa, had the lowest contribution of *GJB2* variants. Moreover, Asia was identified as the highest contributor while the previous report had Europe as the highest contributor of *GJB2* PLP to HI [10]; this can be attributed to the increasing interest and number of genetic researches in all parts of the world. Despite the above differences, the commonly reported PLP *GJB2* variants were similar in both studies. Furthermore, our study and the studies from Chan and Chang and Tsukada reported a similar ethnic-specific spectrum of the common PLP variants in *GJB2* [10,62].

The most common *GJB2* variant is p.Gly12ValfsTer2 (c.35delG) which is frequently reported among populations in Europe, the Middle East, Australia, North, and South America [10]. We observed widespread of this variant across the globe but it was almost absent in sub-Saharan Africa although there were studies from Ghana [59,63], Cameroon [64,65] and South Africa [64] that investigated this variant in African populations. Morocco is an exception, where five independent studies identified biallelic c.35delG mutation in hearing-impaired patients [66,67,68,69,70]. The spread of the variant from Europe and North Africa to North and South America seems to follow migration patterns [10].

Second to *GJB2*: c.35delG is *GJB2*- p.M34T (c.101T > C) which was found to be most prevalent in the United Kingdom (UK). The carrier rate of *GJB2*: p.M34T was calculated at 2.69% in the UK, which was almost twice the carrier rate of *GJB2*-c.35delG (1.36%). In the United States of America, the carrier rate for the *GJB2*: p.M34T variant was found to be 2.3% [71]. The high carrier rates of variants suggested the possibility of heterozygous advantage. However, the audiometric characterization of *GJB2*-p.M34T carriers was not different from homozygous hearing individuals. Hence there is no effect on the hearing ability of the carriers [72].

In the present review, we identified three variants (*GJB2*: p.L79Cfs/c.235delC, p.V37I/c.109G > A, and p.H100RfsTer14/c.299_300delAT) with very high allele frequencies from Asia compared to other continents. These variants were absent in sub-Saharan African countries but were found in a few cases in some North African countries. The Chinese population was found to have a high prevalence of *GJB2*: c.235delC [73] with frequencies of about 14.7% homozygous among a hearing-impaired sub-population, and 16.1% heterozygous in the hearing population [74]. The carrier frequency of *GJB2*: c.235delC is similar to that of the entire Asian population [73,74], and a high prevalence of that variant was reported in Japan [75,76], Korea [77], and Taiwan [78].

The *GJB2*: p.V37I variant was described as a polymorphism by some researchers while others consider it a potential disease-causing missense mutation [79]. The high carrier frequency of the variant among hearing controls informs the polymorphism argument, however individual homozygous of the variant had HI [80]. Compound heterozygosity of the *GJB2*: p.V37I variant and other known *GJB2* pathogenic variants produced mild to severe HI. It was proposed that the milder phenotype was due to the *GJB2-*p.V37I allele [81]. We have identified several independent studies that reported the variant in hearing-impaired individuals, implying that the variant is likely disease-causing. In addition, *GJB2*: p.V37I was predicted as pathogenic by CinVar, Varsome [82], and InterVar [83]. The majority of *GJB2*: p.V37I mutated alleles were identified among Asians and mostly Chinese [79,80]. The third most common *GJB2* variant associated with HI in Asia was c.299_300delAT with an estimated allele frequency of 3.89% [76,84]. Although this variant is very prevalent in China, it appears that this variant is not common in other populations [85].

The truncating *GJB2* mutation p.W24X is the predominant mutation among the Indian and European Gypsy populations [86,87,88]. The *GJB2*: p.W24X was the most commonly observed mutation and accounted for about 95% of all *GJB2* mutations found in the Indian population with a carrier frequency of 2.4% [88]. The mutation was proposed to be a founder effect and confirmed through haplotype analysis of the flanking markers of the *GJB2* gene [86,88].

*GJB2*: c.167delT was reported to be common in the Eurasian populations and postulated to have a single origin of allele due to the observed conserved haplotypes around the mutation [89]. Although the mutation was prevalent in the territories of the Middle East [89,90,91], we found a high number of alleles with PLP in the United States of America. The fourth most common *GJB2* mutation in the American population is *GJB2*: c.167delT and was found to account for about 3.6% of cases. The variant was more prevalent in the White-American population compared to other populations [92].

As, an exception in populations of African ancestry, the *GJB2*: p.R143W is a founder mutation reported first in a Ghanaian village known for its extremely high number of deaf people. To date, *GJB2*: p.R143W mutation is still the most common HI gene in Ghana [59,63,93]. In 1998, there was a study that reported the homozygous form of the mutation in all 11 families investigated [94]. An update from the Ghanaian village in 2020 found 7 out of 8 families with the homozygous mutation, the 8th family had the heterozygous form [93]. The phenotype-genotype correlation from both studies showed that patients with biallelic *GJB2:* p.R143W had profound HI while no difference was observed between the carriers and normal hearing participants [93,94]. In our review, we identified studies from the United States of America and Asia that had reports of patients with the mutated *GJB2:* p.R143W. Considering that the p.R143W mutation has a high frequency in Ghana, it is likely that the mutation emerged in this population and was introduced in the other populations with African ancestry in the diaspora via ancient and/or recent migration events, specially through the Black African transatlantic slave trade.

In our current review, we identified some rare *GJB2* variants that were absent from the databases used. Although these variants were identified in hearing-impaired participants, they may not be considered as HI variants since there are no functional and/or population studies on these variants. A recent report showed that rare variants within the coding region of *GJB2* gene and other HI genes are associated with Meniere disease (MD) [95]. The signs and symptoms of MD are sensorineural hearing loss, tinnitus, and episodic vertigo which are sometimes common to patients with NSHI [96]. Meniere disease was reported to be extremely rare in individuals of African descent which supported the negligible contribution of *GJB2* variants to HI in sub-Saharan Africa [97].

Rare known PLP variants in *GJB2* were found to displace some degree of ethnic specificity in some populations. Similar to previous reports [98,99], we identified extremely high frequencies of *GJB2*: c.IVS1+1G > A from Yakutia and Russia; the high prevalence of the variant within the Yakutia population can be seen in the high carrier rate of 10% [99]. This is also suggestive of a potential accumulation of *GJB2* pathogenic variants in this population and calls for public health attention [98]. The Siberian population which comprises Russia, Kazakhstan, boarders of Mongolia and China was found to have the rare *GJB2*: p.W172C (c.516G > C). We analyzed other variants (p.W172* (c.516G > A), p.W172R (c.514T > A)) at the same site of mutation, and similar to our observation, a previous study reported an extremely high frequency of 62.9% from these populations [100]. Other population-specific *GJB2* rare variants found were p.W44X, p.Q7X, and p.S199F which are common to North America [73,92,101], Ecuador [102], and Colombia [103] respectively.

Although not in high numbers, we identified pathogenic *GJB2* intronic variants that were reported in hearing-impaired patients. These variants were c.IVS1 − 1G > A [104], c.IVS1 − 15C > T [73], c.IVS1 + 12G > A [105], c.IVS1 + 27G > C [106], and c. − 23 + 1G > A. The −23 + 1G > A variant was reported in studies from Russia [107], Poland [108], China [109], India [88], United States of America [71], and with a high prevalence in Turkey [110].

*GJB6* (OMIM:604418) encodes connexin 30, which is part of the family of proteins that form gap junction channels. The location of *GJB6* in the human genome is chromosome 13q12.11 (the same as *GJB2*). The *GJB6* gene is expressed mainly in the brain and skin [111,112] with about 76% protein identity when compared to human *GJB2* [113]. The *GJB6* gene has been associated with Clouston syndrome and hearing impairment. The association of *GJB6* coding region variations to HI has recently been refuted [20]; however, previous reports from Taiwan [23] Germany [24], and Iran [25] identified pathogenic variants within the coding regions of the gene. Six *GJB6* large genomic deletions have been found and previously reported, they are: >920kb deletion [114], 179 kb deletion [115], 131 kb deletion [116], del(*GJB2*-D13S175), del(*GJB6*-D13S1830) and del(*GJB6*-D13S1854) [117]. In addition to the six, there was a report of a seventh *GJB6* deletion, del(*GJB6*-D13S1834), in February 2020 [118]. We identified the large genomic deletions, *GJB6*-D13S1830, and *GJB6*-D18S1854 as the frequently reported *GJB6* variants. In many cases, the large deletions were in trans with pathogenic *GJB2* variants and similar to previous observations [20]. Unlike the *GJB6* coding region variations, the large deletions disrupt a 5′ cis-acting element upstream of both genes. The disruption of the cis-acting element abolishes the expression of *GJB2* and hence is responsible for the phenotype [119,120]. The previously reported *GJB6* knockout mice had significant reduction in the *GJB2* expression which was the cause of HI [121]. Mice models with only the coding region of *GJB6* deleted were found to have normal hearing which provided evidence that the coding region plays no role in the development of HI [115,116].

Our analysis of the global distribution of *GJB6*-D13S1830 confirmed a previous observation by del Castillo et al. [122,123] with a high prevalence of the variant in North America, South America, and Europe with no record from Asia, Australia, and Africa. This variant is known to be frequent in Spain, France, the United Kingdom, Israel, and Brazil [123], and similarly, we observed that the highest number of alleles were from Spain and France. The absence of *GJB6*-D13S1830 from the Asian, Australian, and African populations is indicative of a population-specific spread of the variant and it would inform future studies as well as inform public health policies.

Connexin 43 (*GJA1*, OMIM:121014) is located on chromosome 6 (6q21-q23.2) [50] of the human genome and has been implicated in a number of diseases but mainly in oculodentodigital dysplasia [124,125,126] with pleiotropic phenotypes [127]. It should be noted that numerous pathogenic variants that have been linked to HI were found in the *GJA1* pseudogene, which could explain their lack of representation in HI-associated genes. Indeed, *GJA1* pseudogene has the features of an expressed gene [128]. The messenger RNA of *GJA1* gene was identified in tumor cells which was contrary to the characteristics of pseudogenes (inability to produce functional mRNA and proteins) [129]. Also, *GJA1* is ubiquitously expressed in many human tissues and cells [130]. Although *GJA1* has been associated with HI, only a few pathogenic variants (*GJA1*: p.L11Y, p.V24A, p.L181F, and p.S69P) were reported in deaf patients from the USA [50] and Taiwan [40]. This gene’s contribution to HI is not conclusive considering the low number of patients with the pathogenic variants, and more data from diverse populations are needed to refine the gene-disease pair curation. It is possible that the voltage-gating mechanism of connexin 43 may be affected by the pathogenic variants, culminating in defective gap junction channels. The expression pattern from the mouse genome database has shown that *GJA1* is expressed in the auditory system and may play key roles in hearing and the functioning of the ear [50,131].

The connexin 45 gene, *GJC1* (OMIM:608655)*,* is located on chromosome 17q21.31. Connexin 45 is a candidate HI gene expressed in the auditory system as part of the connexin proteins; however, the mouse genome database has no report of its association with the HI phenotype [131]. Cardiovascular disorders are the most common phenotypes associated with *GJC1* [132]. We found a research effort to identify gene variants in *GJC1* among hearing-impaired participants [52]. The authors studied participants from 3 different populations but did not find any pathogenic variants from the HI cohorts studied. It is imperative to screen a larger population for *GJC1* mutations to investigate the gene’s contribution to HI.

In humans, *GJB3* (connexin 31 gene, OMIM:603324) found on chromosome 1p34.3, encodes gap junction beta 3 (protein [133]. The gene has been associated with two major conditions erythrokeratodermia variabilis et progressive (MIM:133200) and non-syndromic hearing loss. In our review, we found only a few hearing-impaired participants with mutations in *GJB3*, at low allele frequencies. However, only two likely pathogenic variants (*GJB3*-p.E183K/c.547G > A/rs74315318 and *GJB3*-p.N166S/c.497A > G/rs121908851) were reported from publications considered [134]. We could not conclude on the clinical significance of these variants since the 3 databases gave conflicting clinical significance [82,83,135]. Functional analysis suggested an expression overlap and possible interactions between connexin 26 and connexin 31 in the cells at the tip of the spiral limbus. It was further demonstrated that the two connexins formed heterotypic channels in the [38]. The data from the above functional studies confirmed a digenic form of NSHI. In addition, a biallelic mutation in any of the two connexins can result in NSHI. Confirming the digenic claim, *GJB2* p.V37I/p.L213S/*GJB3* p.V84I was found to co-segregate with NSHI in a family [42]. To fully understand the role of *GJB3* in HI, there is a need to study many populations as well as conduct further functional assays to assess the ironic and biochemical functions of the gene and its mutant forms.

Like other connexins, *GJB4* (OMIM:605425) encodes connexin 30.3 which oligomerizes to form gap junction channels. Unlike the common connexins, the contribution of *GJB4* to HI remains unknown. There were considerable efforts by five groups of researchers identified in this review to investigate *GJB4* mutations in deaf populations. The clinical significance of the identified variants of *GJB4* had conflicting clinical significance; six variants were predicted as pathogenic or likely pathogenic on not more than one of the three databases used [82,83,135]. Functional genomics using mouse models showed that the auditory system of *GJB4* null mice is unaffected, the mutant mice had normal hearing when assessed by brain stem-evoked potentials [136]. However, the gene was found to be expressed in the mouse auditory system [131] and rat cochlear [137] suggesting its role in hearing.

The association of *GJC3* (a gene that encodes connexin 29, OMIM:611925) to NSHI remains unclear although a few studies have associated variations in connexin 29 to NSHI. These variants as presented in our results were predicted to have uncertain clinical significance. However, data from the Mouse Genome Informatics has shown that *GJC3* is expressed in the auditory system and has hearing loss as one of its phenotypes [131]. Functional studies using *GJC3* null mice showed reduced maturation of the hearing threshold. Noise-induced HI has been reported in *GJC3* null mice [131], while another study conducted with adult mice did not find any difference in the hearing threshold of *GJC3* null and wildtype mice [138]. To conclude on the contribution of *GJC3* to HI, we recommend screening more deaf populations across the globe.

There was a limited number of publications retrieved on *GJB3*, *GJB4*, *GJC3, GJA1*, and *GJC1* which was a major challenge encountered in this study. Therefore, it was difficult to determine whether these genes should be considered or catalogued as HI genes. Apart from *GJB2,* which is globally known to be associated with HI, there is a need for more studies from different populations as well as functional studies for a concluding decision to be made on *GJB4*, *GJA1, GJC3*, and *GJC1.* The Hearing Loss Home Page [8] has categorized *GJB3* as an autosomal dominant HI gene. Although *GJB6* was previously considered as a HI gene, recent studies have disproved its classification as a HI gene. The present review suggests that only *GJB2* and *GJB3* are recognized and validated HI genes.

## 4. Materials and Methods

### 4.1. Search Terms

We reviewed publications on connexin genes variants implicated in human HI and the distributions of the common variants across the globe. Also, we evaluated the methods used to investigate NSHI-implicated gene variants. The protocol was registered on PROSPERO, International Prospective Register of Systematic Reviews with the registration number “CRD42020169697”. Two independent reviewers conducted the literature search on PubMed, Scopus, Africa-Wide Information, and Web of Science databases. The search term used in the study comprised of three major components: the first component was (connexins OR “gap junction alpha protein” OR “Gap junction alpha proteins” OR “Gap junction beta-protein” OR “Gap junction proteins” OR connexin OR GJB OR GJA), the second component was (“hearing loss” OR “non-syndromic sensorineural hearing loss” OR “non-syndromic deafness” OR “non-syndromic hearing impairment” OR “hearing impairment” OR deafness) and the last component was (genetic OR gene OR genes) OR (“genetic loci” OR genes OR “genetic diseases” OR “genetic markers). Each component of the search term was joined with “AND” to obtain the resultant search term which was used to retrieve publications from the databases used (Figure 1).

### 4.2. Data Extraction

Two independent reviewers conducted the literature search between 1 April to 31 May 2020 and 2592 full-length articles were selected based on the inclusion and exclusion criteria outlined below.

Inclusion criteria:Publications on human hearing impairment;Publications on the genetics of non-syndromic hearing impairment;Publications reporting on connexins association with NSHI.

Exclusion criteria:Studies that are not on human hearing impairment;Review or meta-analysis publications;Policy papers;Publications that are not on connexin hearing impairment;Publications on environmental and/or syndromic hearing impairment;Publications focusing on in silico analysis.

The search with the keywords gave 874, 992, 31, and 695 results from PubMed, Scopus, Africa-Wide Information, and Web of Science databases, respectively. A blinded screening was undertaken by the two reviewers (SMA, and EWT, first and second authors of this manuscript) using the titles as the first-level screening followed by the abstracts. The search results were downloaded into EndNote referencing software and duplicates removed (Figure 6). The data extraction was conducted independently by the two reviewers and compared to remove any form of bias. The following data elements were extracted: (1) location and date; (2) connexin genes investigated; (3) the number of mutant alleles; (4) methods for genetic screening. The data extracted were manually captured onto Microsoft Excel sheets and analyzed using Microsoft Excel (Office 365 education license under the University of Cape Town, South Africa) and SPSS version 25 (IBM, Armonk, New York, United States). A third person (AW) who is an expert in the field was consulted in times of disagreement between the individual judgments during the screening and data extraction process.

### 4.3. Quality Assessment

To avoid any form of bias, two independent reviewers (SMA and EWT) synthesized the data and assessed the quality of the documents included, using the quality of genetic studies (Q-Genie) tool developed by Sohani et al. [139] for genetic studies and the risk of bias assessment tool for prevalence studies developed by Hoy et al. [140] for the other studies. Discrepancies were solved by discussion and consensus. An expert (AW) was consulted to resolve disagreements between the judgment of the reviewers, by discussion and consensus. The quality assessment was conducted at the outcome level of each study. The studies were assessed for selective outcome reporting and whenever there was evidence of this, the effect of the selective outcome reporting on the study results was further analyzed.

### 4.4. Clinical Significance

The clinical significance of the identified variants was assessed on three databases InterVar [83], VarSome [82], and ClinVar [135]. Both VarSome and InterVar are bioinformatic web-based tools build on the American College of Medical Genetics and Genomics (ACMG)/Association for Molecular Pathology (AMP) 2015 guidelines and are useful for clinical interpretation of human genetic variants [82,83]. ClinVar is also a web-based database that provides evidence and relationships between human variants (found in biological sample) and phenotypes which serve as strong evidence for clinical significance interpretation [135]. Our judgement on the clinical significance of each variant was based on the prediction from the 3 databases mentioned above.

## 5. Conclusions

The present comprehensive review on the contribution of connexins genes in HI, globally, found most investigations performed in populations from China, with relatively few studies from African populations. In most populations, except Africans, common *GJB2* pathogenic variants that were found were p.Gly12ValfsTer2 and p.M34T (commonly among Europeans), p.L79Cfs, p.V37I, and p.H100RfsTer14 (mostly in Asians), p.W24X among Indians, p.L56Rfs among Americans, and p.R143W particularly in Ghanaians, the African exception for *GJB2* variants. These *GJB2* variants exhibited population-specific prevalence due to founder effects. The second most common HI-associated connexin was *GJB6.* We identified two main deletions in *GJB6* (*GJB6*-D13S1830 and *GJB6*-D13S1854), that were predicted to be pathogenic, however, the coding region variants of *GJB6* are no longer considered as causes of HI. From the review, we identified 11 *GJA1* variants of which 3 were predicted to be pathogenic but their pathogenicity needs to be confirmed with more data from multiple populations. The *GJB4* variants found from the reports that were used for this review mostly had conflicting clinical significance, but the majority were predicted pathogenic by one of the 3 databases used. None of the *GJC1*, *GJB3,* and *GJC3* variants were predicted pathogenic. Most researchers used targeted sequencing approaches to investigate connexin genes associated with HI. The present review suggests that only *GJB2* and *GJB3* are recognized and validated HI genes. It is likely that the wide use of whole-exome sequencing, particularly in understudied African populations, will rapidly increase the identification of novel HI-associated gene variants and improve disease-gene pairs curation, globally.

## Figures and Tables

**Figure 1 life-10-00258-f001:**
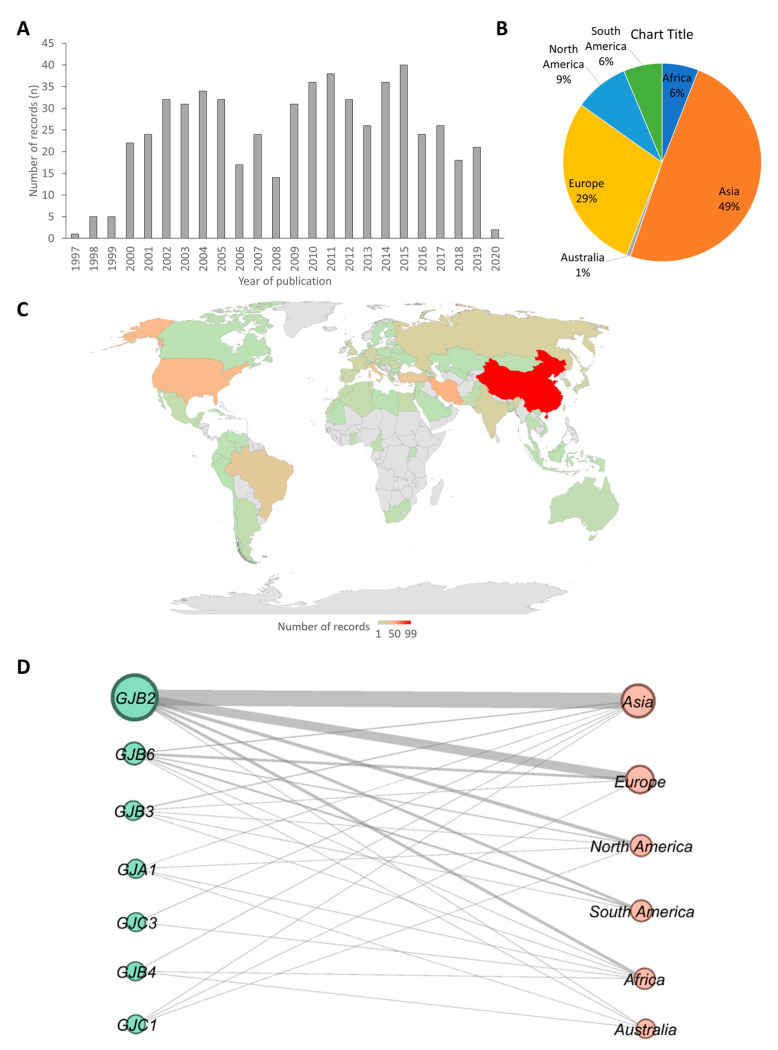
Geographical distributions of the studies included in this review. (**A**) A bar chart showing frequency of articles by the year of publication. (**B**) A pie chart of distribution of articles from which data were extracted by continent. (**C**) A map of countries showing the number studies that reported at least one connexin gene variant. The gray regions have no record included in this study. Different shades of blue were used to represent the number of studies retrieved and reviewed per country with the darkest shade of blue as the highest number and the lightest as the smallest number. The number written on the map denotes the number of studies. The map was created in Microsoft Excel (Office 365 education license under the University of Cape Town, South Africa) (**D**) Network of connexin gene plotted against continents from which they were reported. The nodes on the left (green) and the right (pink) correspond to connexin genes and continents respectively. The size of the nodes and the thickness of the lines between nodes are proportional to the number of publications. The network was built using the open-source software Gephi [22]

**Figure 2 life-10-00258-f002:**
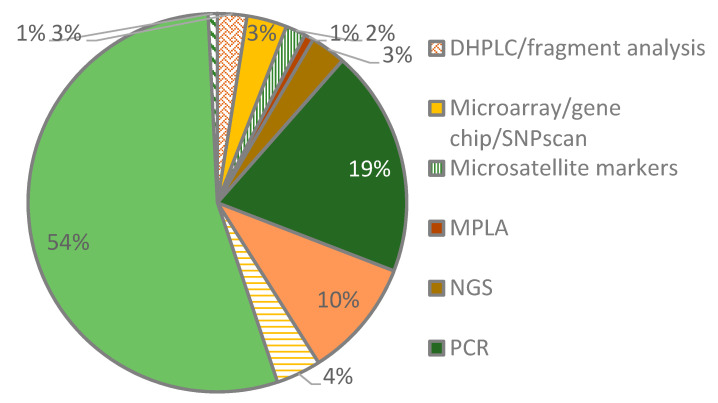
Methods used to investigate connexin gene variants. Among the methods are denaturing high-performance liquid chromatography (DHPLC), multiplex ligation-dependent probe amplification (MLPA), polymerase chain reaction (PCR), next-generation sequencing (NGS), restriction fragment length polymorphism (RFLP), and single-strand conformational polymorphism (SSCP).

**Figure 3 life-10-00258-f003:**
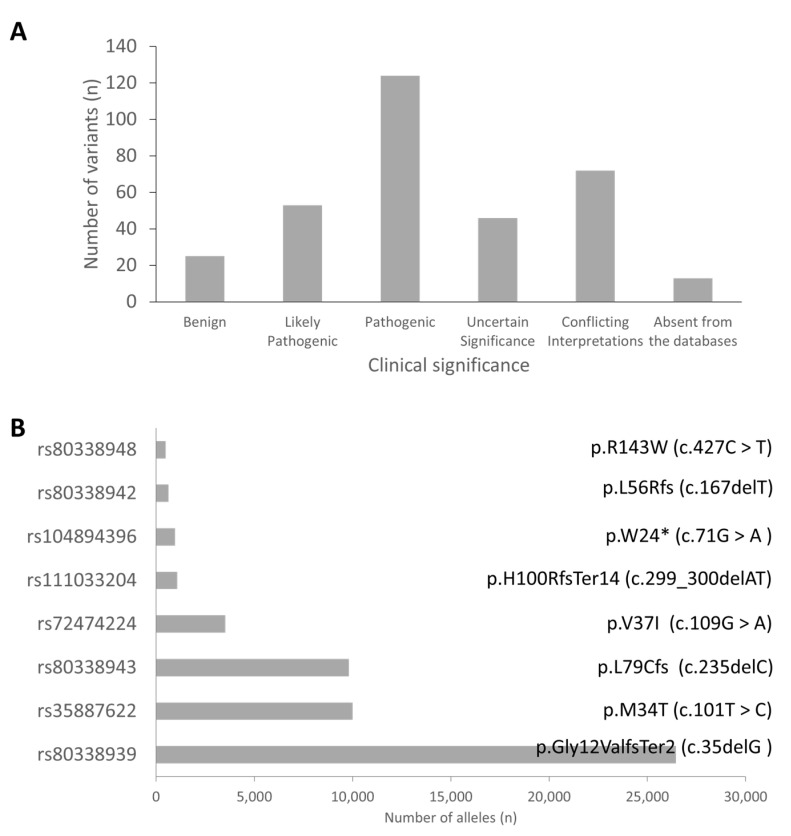
Common *GJB2* variants. (**A**) Clinical significance of identified variants. (**B**) The top eight *GJB2* variants ranked based on the total number of alleles.

**Figure 4 life-10-00258-f004:**
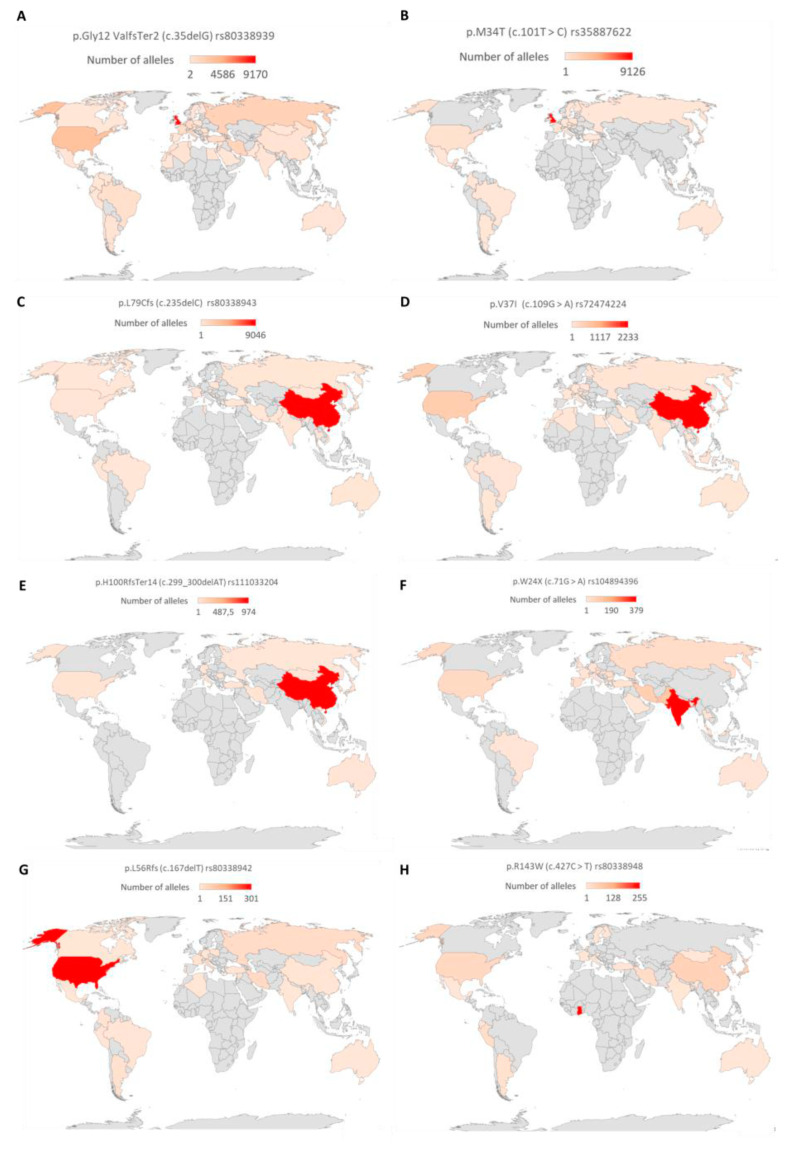
Global distribution of common *GJB2* variants. A graph showing the total number of reported alleles of (**A**) p.Gly12ValfsTer2 (c.35delG), (**B**) p.M34T (c.101T > C), (**C**) p.L79Cfs/c.235delC, (**D**) p.V37I/c.109G > A, (**E**) p.H100RfsTer14/c.299_300delAT, (**F**) p.W24X/c.71G > A, (**G**) p.L56Rfs/c.167delT, and (**H**) p.R143W/c.427C > T. The countries were colored with a gradient from red (highest number of alleles) to brown (lowest number of alleles). Countries shaded grey either had no reports or no alleles. The map was created by the authors in Microsoft Excel (Office 365 education license of the University of Cape Town, South Africa).

**Figure 5 life-10-00258-f005:**
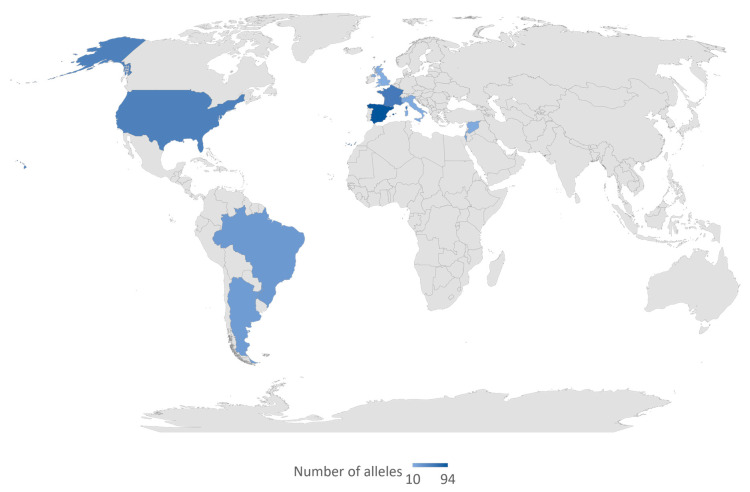
Global distribution of del(G*JB6*-D13S1830). The variant del(*GJB6*-D13S1830) was reported in all the countries highlighted in blue color. The intensity of the blue color denotes the frequency of reported alleles. The map was created by the authors in Microsoft Excel (Office 365 education license of the University of Cape Town, South Africa).

**Figure 6 life-10-00258-f006:**
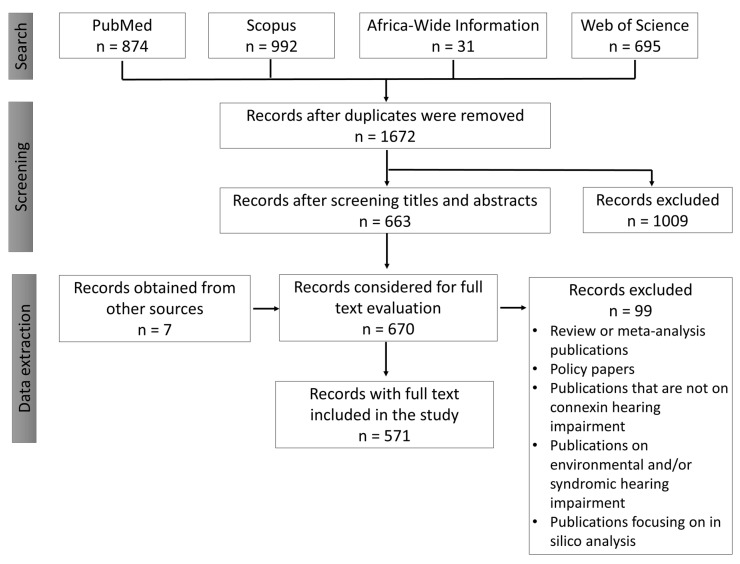
Flow diagram illustrating the screening of articles obtained after the literature search.

**Table 1 life-10-00258-t001:** Global distribution of connexin 30 (GJB6) gene variants.

Country/Territory	Number of Alleles *	Protein Change	Nucleotide Change	Reference Number	Clinical Significance				Reference
					Intervar	Varsome	ClinVar	Verdict	
Taiwan	1/520	p.A40V	c.119C > T	rs780320724	Likely Pathogenic	Likely Pathogenic	Pathogenic	Pathogenic	[23]
Malaysia	3/NA	-	366delT	-	-	-	-	-	[26]
Germany	1/376	-	682insA	-	-	-	-	-	[27]
Uganda	2/230	p.N113K	c.339T > A	rs143766955	Benign	Likely Benign	Benign	Benign	[28]
Uganda	1/230	c.476A > G	p.N159S	rs35277762	Benign	Likely Benign	Benign	Benign	[28]
Malaysia	2	p.E101K	c.301G > A	rs571454176	Likely Benign	Uncertain Significance	Uncertain Significance	Uncertain Significance	[26]
Malaysia	1/NA	p.A148D	c.443_444 delC AinsAC	-	-	Uncertain Significance	-	Uncertain Significance	[26]
Malaysia	1/NA	p.Q124H	-	-	Uncertain Significance	Uncertain Significance	-	Uncertain Significance	[26]
Slovenia	1/144	p.M203V	c.607A > G	rs200674715	Uncertain Significance	Likely Benign	Benign	Benign	[29]
Germany	1/376								[27]
Korea	1/394	p.I248V	c.742A > G	rs747371119	Uncertain Significance	Uncertain Significance	Uncertain Significance	Uncertain Significance	[30]
Qatar	1/NA	p.P70L	c.209C > T	rs727505123	Uncertain Significance	Uncertain Significance	Uncertain Significance	Uncertain Significance	[31]
Korea	1/394	p.P87P	c.261A > T	rs777309137	Likely Benign	Likely Benign	-	Benign	[30]
Germany	6/396	p.T5M	c.14C > T	rs104894414	Likely Pathogenic	Uncertain Significance	Pathogenic	Pathogenic	[24]
Malaysia	1/NA	p.R32Q	c.95G > A	rs766604251	Uncertain Significance	Uncertain Significance	-	Uncertain Significance	[26]
Germany	1/376	p.V190A	c.569 T > C	rs780513857	Uncertain Significance	Uncertain Significance	Uncertain Significance	Uncertain Significance	[26,27]
Malaysia	1/NA	p.I145H	c.433_434 delA TinsCA	-		Uncertain Significance	-	Uncertain Significance	[26]

* The numerators in this column represent the number of mutated alleles, and the denominators the total number of screened alleles. NA, not applicable (the authors were not clear on the total number of alleles they have screened), InterVar, VarSome, and ClinVar are databases to assess the clinical significance of the variants.

**Table 2 life-10-00258-t002:** Global distribution of connexin 31 (*GJB3*) gene variants.

Country/Territory	Number of Alleles	Protein	Nucleotide Change	rs Number	Clinical Significance				Reference
					Intervar	Varsome	ClinVar	Verdit	
Germany	2/376	p.K56Q	c.166 A > C	rs746219527	Uncertain Significance	Likely Benign	Uncertain Significance	Uncertain Significance	[27]
Germany	1/376	p.R101Q	c.302G > A	rs765605645	Uncertain Significance	Likely Benign	-	Conflicting Interpretations	[27]
Germany	1/376	p.R106H	c.317 G > A	rs369979083	Uncertain Significance	Uncertain Significance	Likely Benign	Uncertain Significance	[27]
Tunisia	1/NA	p.R32W	c.94C > T	rs1805063	Benign	Benign	Likely Benign	Benign	[32]
Austria	2/90								[33]
USA	2/126								[34]
Brazil	2/NA								[35]
Tunisia	4/NA	p.N119N	c.357C > T	rs41310442	Benign	Benign	Benign	Benign	[32]
Austria	4/90								[33]
China	7/186								[36]
USA	1/126								[34]
Morocco	1/390								[37]
Korea	36/424								[30]
China	1/216	p.N166S	c.497A > G	rs121908851	Uncertain Significance	Likely Benign	Pathogenic	Conflicting Interpretations	[38]
Korea	1/20	p.A194T	c.580G > A	rs117385606	Benign	Benign	Benign	Benign	[39]
China	2/216								[38]
Korea	7/430								[30]
Taiwan	4/506								[40]
China	2/NA								[41]
China	3/206	p.V84I	c.250G > A	rs145751680	Benign	Benign	Benign	Benign	[42]
Korea	7/424								[30]
Taiwan	1/506								[40]
Korea	1/40								[43]
China	1/NA								[41]
Austria	11/90	p.N266N	c.798C > T	rs35983826	Benign	Benign	Benign	Benign	[33]
USA	10/126								[34]
China	4/186								[36]
China	12/170								[41]
China	2/186	p.S11S	c.33C > T	rs112499125	Likely Benign	Benign	Likely Benign	Benign	[36]
USA	2/126	p.N67N	c.201C > T	-	Likely Benign	Uncertain Significance	-	Conflicting Interpretations	[34]
Korea	1/424	pV27M	c.79G > A	rs775072109	Uncertain Significance	Benign	-	Conflicting Interpretations	[30]
Korea	1/424	p.V43M	c.127G > A	rs761320902	Uncertain Significance	Likely Benign	-	Conflicting Interpretations	[30]
Korea	415/430		c.813+43C > A	rs41266429	-	Benign	Benign	Benign	[30]
Korea	351/430		c.813+53G > A	rs476220	-	Benign	Benign	Benign	[30]
China	2/4	p.E183K	c.547G > A	rs74315318	Likely Pathogenic	Benign	Conflicting Interpretations	Conflicting Interpretations	[44]
Taiwan	1/506								[40]
China	2/4	p.R180 *	c.538C > T	rs74315319	Uncertain Significance	Benign	Uncertain Significance	Uncertain Significance	[44]
Taiwan	2/506	p.L10R	c.29T > G	-	Uncertain Significance	Uncertain Significance	-	Uncertain Significance	[40]
Taiwan	1/506	p.T18I	c.53C > T	rs755025684	Uncertain Significance	Benign	-	Conflicting Interpretations	[40]
Brazil	1/NA	p.49delK	c.1227C > T	-	-	-	-	-	[35]
Australia	3/520	p.V174M	c.520G > A	rs749431664	Uncertain Significance	Uncertain Significance	-	Uncertain Significance	[40]
Brazil	2/414	p.Y177D	c.529T > G	rs80297119	Benign	Benign	Benign	Benign	[45]
Brazil	2/4								[35]
China	2/186	p.G256S	c.766G > A	-	Likely benign	Uncertain significance		Conflicting Interpretations	[36]

* The numerators in this column represent the number of mutated alleles, and the denominators the total number of screened alleles. NA, not applicable (the authors were not clear on the total number of alleles they have screened), InterVar, VarSome, and ClinVar are databases to assess the clinical significance of the variants.

**Table 3 life-10-00258-t003:** Connexin 30.3 (*GJB4*) gene variants.

			Clinical Significance				Ghana	Australia	Iran	China
Protein Change	Nucleotide Change	rs Number	Intervar	Varsome	InterVar	Verdit	[21]	[23]	[46]	[15]
p.C169 *	c.507C > A	rs79193415	Uncertain Significance	Pathogenic	-	Conflicting Interpretations	-	2/NA	-	1/506
p.C169C	c.507C > T	rs79193416	Likely Benign	Uncertain Significance	-	Conflicting Interpretations	-	-	2/144	
p.E67L	c.199G > A	rs368331423	Uncertain Significance	Benign	-	Conflicting Interpretations	-	-	-	1/506
p.G126T	c.376G > A	rs146979528	Likely Pathogenic	Benign	-	Conflicting Interpretations	-	-	-	2/506
p.H221Y	c.661C > T	rs1223189096	Uncertain Significance	Likely Benign	-	Conflicting Interpretations	-	-	-	1/506
p.R101H	c.302G > A	rs375702737	Likely Pathogenic	Likely Benign	-	Conflicting Interpretations	-	1/520	-	-
p.R103C	c.307C > T	rs9426009	Benign	Benign	-	Benign	-	-	1/144	-
p.R124W	c.370C > T	rs373126632	Likely Pathogenic	Benign	-	Conflicting Interpretations	-	1/520	-	1/506
p.R227W	c.679C > T	rs185327282	Uncertain Significance	Likely Benign	-	Conflicting Interpretations	-	-	1/144	-
p.R22C	c.64C > T	rs776245625	Likely Pathogenic	Likely Benign	-	Conflicting Interpretations	-	1/520	-	1/506
p.R98C	c.292C > T	rs200602523	Likely Pathogenic	Benign	-	Conflicting Interpretations	-	2/520	-	1/506
p.T233L	c.698C > A	-	Uncertain Significance	Likely Benign		Conflicting Interpretations	-	-	-	1/506
p.V37M	c.109G > A	rs146378222	Benign	Benign	Uncertain Significance	Benign	-	2/520	-	2/506
p.V74M	c.220G > A	rs771048190	Likely Pathogenic	Likely Benign	-	Conflicting Interpretations	-	1/520	-	-
p.N119T	c.356A > C	rs190460237	Likely Pathogenic	Uncertain Significance	-	Conflicting Interpretations	2/400	-	-	-
p.E204A	611A > C	rs3738346	Benign	Benign	Benign	Benign	70/400	-	-	-
p.R151S	c.451C > A	rs78499418	Benign	Benign	-	Benign	58/400	-	-	-
p.T172T	c.516T > C	rs111693060	Benign	Benign	Benign	Benign	13/400	-	-	-
p.K123K	c.369G > A	rs142843509	Likely Benign	Benign	Likely Benign	Benign	2/400	-	-	-
p.R101R	c.303C > G	rs138184343	Likely Benign	Benign	Benign	Benign	15/400	-	-	-
p.Q80*	c.238C > T	rs114429815	Benign	Benign	Benign	Benign	3/400	-	-	-

* The numerators in this column represent the number of mutated alleles, and the denominators the total number of screened alleles. NA, not applicable (the authors were not clear on the total number of alleles they have screened), InterVar, VarSome, and ClinVar are databases to assess the clinical significance of the variants.

**Table 4 life-10-00258-t004:** Connexin 29 (*GJC3*) gene variants.

							* Number of Alleles				
							Ghana	Taiwan	India	China	
Protein Change	Nucleotide Change	rs Number	Intervar	Varsome	ClinVar	Verdict	[21,47]	[40]	[47]	[48]	
p.I190N	c.569T > A	rs121908693	Uncertain Significance	Uncertain Significance	-	Uncertain Significance			1/246		
p.R15G	c.43C > G	-	Uncertain Significance	Uncertain Significance	-	Uncertain Significance				1/506	
p.W77S	c.230G > C	-	Uncertain Significance	Uncertain Significance	-	Uncertain Significance				1/506	
p.L17S	c.525T > G	rs752804324	Likely benign	Likely benign		Likely benign				2/506	
-	c.781 + 62G > A	rs116853822	-	Likely Benign	-	Likely Benign		10/520		8/506	
p.M1R	c.2T > G	-	Uncertain Significance	Uncertain Significance	-	Uncertain Significance		3/520		3/506	
p.E269D	c.807A > T	rs763649019	Uncertain Significance	Likely Benign	Uncertain Significance	Uncertain Significance		1/520			
p.P164S	c.490C > T	rs73405465	Benign	Benign	-	Benign	53/400				

* The numerators in this column represent the number of mutated alleles, and the denominators the total number of screened alleles, InterVar, VarSome, and ClinVar are databases to assess the clinical significance of the variants.

**Table 5 life-10-00258-t005:** Connexin 43 (*GJA1*) gene variants.

Country/Territory	* Number of Alleles	Protein	Nucleotide Change	rs Number	Clinical Significance				Reference
					Intervar	Varsome	ClinVar	Verdict	
Taiwan	1/520	p.S69P	c.205T > C	-	Likely pathogenic	Likely pathogenic	-	pathogenic	[23]
Taiwan	16/520	-	c.932delC	-	-	-	-	-	[23]
Taiwan	2/520	-	c.976C > T	-	-	-	-	-	[23]
Taiwan	1/506	p.L181F	c.543G > C	-	Likely pathogenic	Likely pathogenic	-	pathogenic	[40]
Cameroon	2/134	-	c.-16-51A > G	rs189167598	-	Benign	-	Benign	[49]
South Africa	1/46	p.N63N	c.189T > C	rs139688042	Likely benign	Uncertain Significance	-	Conflicting Interpretations	[49]
South Africa	1/46	p.N122N	c.366T > C	-	Likely benign	Uncertain Significance	-	Conflicting Interpretations	[49]
Cameroon	11/134	p.R239R	c.717G > A	rs57946868	Benign	Uncertain Significance	-	Conflicting Interpretations	[49]
South Africa	2/46								
South Africa	1/46	p.A253V	c.758C > T	rs17653265	Benign	Benign	Benign	Benign	[49]
USA	6/52	p.L11Y	c.31–32 delCTinsTA	-	-	Likely pathogenic	-	Likely pathogenic	[50]
USA	2/20	p.V24A	c.71T > C	-	Likely pathogenic	Likely pathogenic	-	pathogenic	[50]

* The numerators in this column represent the number of mutated alleles, and the denominators the total number of screened alleles, InterVar, VarSome, and ClinVar are databases to assess the clinical significance of the variants.

**Table 6 life-10-00258-t006:** Connexin 45 (*GJC1*) gene variants.

Country	USA		Turkey	UK
Reference	[52]		[52]	[52]
Protein change	p.L71L	p.T302T	p.D297N	L304L
Nucleotide change	c.213C > T	c.906C > T	c.889G > A	c.912G > T
rs number	rs61749924	rs2229395	-	-
* Number of alleles	2/168	13/120	4/194	1/80
Intervar	Likely benign	Likely benign	Uncertain significance	Likely benign
Varsome	Uncertain significance	Uncertain significance	Uncertain significance	Uncertain significance
ClinVar	-	-	-	-
Verdict	Conflicting Interpretations	Conflicting Interpretations	Conflicting Interpretations	Conflicting Interpretations

* The numerators in this column represent the number of mutated alleles, and the denominators the total number of screened alleles, InterVar, VarSome, and ClinVar are databases to assess the clinical significance of the variants.

**Table 7 life-10-00258-t007:** Summary of the global allele frequencies of the common pathogenic (PLP) variants in connexin genes associated to hearing impairment (HI).

		* Patients (#Chrom/Total #Chrom (Allele Frequency))						* Controls (#Chrom/Total #Chrom (Allele Frequency))					
Gene	Variant	Africa	Asia	Australia	Europe	North America	South America	Africa	Asia	Australia	Europe	North America	South America
* GJB2 *	p.Gly12 ValfsTer2 (c.35delG)	770/3848 (20.0%)	5917/71,209 (8.3%)	50/104 (48.1%)	15,616/65,019 (24.0%)	3237/26,976 (12.0%)	822/8191 (10.0%)	12/1604 (0.7%)	35/4684 (0.7%)		197/21,978 (0.9%)	6/988 (0.6%)	
	p.M34T (c.101T > C)	2/272 (0.7%)	89/4902 (1.8%)	5/104 (4.8%)	9418/47,909 (19.7%)	451/25,118 (1.8%)	44/4500 (1.0%)				13/1886 (0.7%)	5/588 (0.9%)	
	p.L79Cfs (c.235delC)	2/262 (0.8%)	9666/250,680 (3.9%)	31/520 (6.0%)	32/4382 (0.7%)	80/20,406 (0.4%)	2/866 (0.2%)		35/4908 (0.7%)		1/1886 (0.1%)	1/588 (0.1%)	
	p.V37I (c.109G > A)	24/1192 (2.0%)	2833/81,139 (3.5%)	8/104 (7.7%)	95/13,227 (0.7%)	530/27,288 (1.9%)	30/4852 (0.6%)	3/640 (0.5%)	200/3478 (5.8%)			11/588 (1.9%)	
	p.H100Rfs Ter14 (c.299_300delAT)	0/0	1046/85,332 (1.2%)	7/520 (1.3%)	7/2936 (0.2%)	20/19,067 (0.1%)	0/0		3/1264 (0.2%)				
	p.W24X (c.71G > A)	0/0	666/22,464 (3.0%)	3/104 (2.9%)	249/12,523 (2.0%)	47/17,055 (0.3%)	7/2248 (0.3%)		11/320 (3.4%)				
	p.L56Rfs (c.167delT)	2/50 (4.0%)	240/17,350 (1.4%)	1/104 (1.0%)	93/12,141 (0.8%)	275/21,540 (1.3%)	30/3094 (1.0%)		7/2690 (0.3%)		1/1886 (0.1%)		
	p.R143W (c.427C > T)	255/1298 (19.6%)	154/62,605 (0.2%)	1/104 (1.0%)	17/2977 (0.6%)	35/21,189 (0.2%)	35/2132 (1.6%)	2/290 (0.7%)					
* GJB6 *	Del (GJB6- D13S1830)	1/204 (0.5%)	31/3096 (0.1%)	2/68 (2.9%)	186/7778 (2.4%)	36/1498 (2.4%)	44/4516 (1.0%)	0/198	0/782		0/1502	0/230	0/1508
	Del (GJB6- D13S1854)				1/782 (0.1%)		10/2524 (0.4%)						
* GJA1 *	p.L11Y (c.31–32 delCTinsTA)				6/52 (11.5%)						0/200		
	p.V24A (c.71T > C)				2/20 (10.0%)						0/200		
	p.L181F (c.543G > C)		1/506 (0.2%)						0/240				
	p.S69P (c.205T > C)		1/520 (0.2%)						0/240				

* The numerators in this column represent the number of mutated alleles, and the denominators the total number of screened alleles. #chrom = number of chromosomes, InterVar, VarSome, and ClinVar are databases to assess the clinical significance of the variants.

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
