# Peer review of "Connexin Genes Variants Associated with Non-Syndromic Hearing Impairment: A Systematic Review of the Global Burden"

_life, 2020, doi:10.3390/life10110258_

Round 1

Reviewer 1 Report

This is a comprehensive study by Adadey looking at the prevalence of Cx mutations associated with hearing at the global level. The current study focuses on the different mutations associated with seven different connexin subtypes that have either been identified or suspected to cause hearing loss in the human population. The authours show the common geographical locations of each mutation as well as their prevalence and potential pathogenicity. Similar to other studies, this study concludes that mutations in GJB2 were the most prevalent in hearing impaired subjects. They also find GJB6 mutations/deletion the second most prevalent. Overall, while the study was well-designed, I have a few recommendations to improve the manuscript outlined below.

In discussing the methods used for sequencing by the papers outlined in this study, the authours conclude that targeted sequencing was the most common method used. Were there common primer sets that were used amongst studies or were they all different? It would be useful to have a table list of the primers that were used for sequencing in the supplemental data for comparison between studies.

When talking about the GJA1I gene (encoding Cx43) mutations, it would be beneficial to stress that many of the mutations that have been linked to hearing loss were found in the GJA1 pseudogene which could explain their lack of representation of genes involved in hearing impairment. This was not stated at all and is very important in terms of Cx43 and hearing.

The authours mention the chromosome location of many, but not all, of the chromosome locations of the connexin genes. For example, it is mentioned in the manuscript that Cx30 is found on chromosome 13 but it is not mentioned where Cx26 is located. It would be useful to mention that Cx30 and Cx26 are close together which would strengthen the argument for Cx30 not being involved in hearing impairment as much.

With some connexins the authours mention the localization and function of the subtypes in hearing but other are left out. I would recommend that the localization in the auditory tract for all seven connexins examined in the study be defined in the study. Since this study is examining human patients, it would be advisable to also include the human localization studies of which many of these connexin subtypes have been examined in terms of localization in the ear.

Finally, it would be valuable to have a final figure in the main manuscript summarizing whether each connexin is important in hearing or not. For some subtypes the authours mention that there are confounding reports. However, it is not clear to me what the final conclusions are after performing this comprehensive analysis. A final figure/table/schematic outlining each connexin subtype and their relation to hearing impairment would complement the findings of the study.

Minor:

1) Line 51-53: This is a confusing sentence, consider rewording

2) Line 53: Change “distinguish” to “distinguishing”

3) Line 91: [1] is in a different reference format then the rest of the introduction

4) Line 108: Change “extract” to “extracted”

5) Figure 1C: The number of records is small and hard to see. The different shades of blue are also difficult to differentiate

6) Line 118: Change “compare” to “compared”

7) Line 119: Change “ continent” to “continents”

8) Line 116: In figure 4 legend. Should read “global distribution of common…”

9) All tables are cut off in PDF so cannot see the first and last of it. Consider addressing.

10) Line 219: Change “was” to “were”

11) In the discussion the style of references has changed. This needs to be consistent throughout. 

12) Line 276: Change “migrations” to “migration”

13) Line 327: Change to Introduced

14) Line 334: There is a “.” After gene. Get rid of the word “impaired”

15) Line 338: Should read “and hence is responsible…”

16) Line 339: Change “mice” to “mouse”

17) Line 339: Should mention here that the original Cx30 knock-out mouse also reduced Cx26 expression and that was why they had hearing loss. Tuebner et al. 2003, Connexin 30 deficiency causes severe impairment and lack of endocochlear potential, Genetics should be cited.

18) Line 354: Change to “It is possible…”

19) Line 356: Need a reference here, what evidence is there for this theory?

20) Line 358: “Encodes” is misplaced, revise.

21) Line 403: “also” is written twice, revise.

22) Line 440: Change “unto” to “onto”.

Author Response

Reviewer 1

This is a comprehensive study by Adadey looking at the prevalence of Cx mutations associated with hearing at the global level. The current study focuses on the different mutations associated with seven different connexin subtypes that have either been identified or suspected to cause hearing loss in the human population. The authours show the common geographical locations of each mutation as well as their prevalence and potential pathogenicity. Similar to other studies, this study concludes that mutations in GJB2 were the most prevalent in hearing impaired subjects. They also find GJB6 mutations/deletion the second most prevalent. Overall, while the study was well-designed, I have a few recommendations to improve the manuscript outlined below.

Response:  We are grateful you for the positive comments.

Comment: In discussing the methods used for sequencing by the papers outlined in this study, the authours conclude that targeted sequencing was the most common method used. Were there common primer sets that were used amongst studies or were they all different? It would be useful to have a table list of the primers that were used for sequencing in the supplemental data for comparison between studies.

Response: The list of primers is provided in the supplementary material, and the line below added to the results section:  

Line 125: The analysis of the primers sets used for sequencing revealed their uniqueness for each study (Table S1).

Comment: When talking about the GJA1I gene (encoding Cx43) mutations, it would be beneficial to stress that many of the mutations that have been linked to hearing loss were found in the GJA1 pseudogene which could explain their lack of representation of genes involved in hearing impairment. This was not stated at all and is very important in terms of Cx43 and hearing.

Response: Thanks for this important suggestion. The discussion has been updated accordingly, as follows:

Line 414: It should be noted that numerous pathogenic variants that have been linked to HI were found in the GJA1 pseudogene, which could explain their lack of representation in HI-associated genes. Indeed, GJA1 pseudogene has the features of an expressed gene [132]. The messenger RNA of GJA1 gene was identified in tumor cells which was contrary to the characteristics of pseudogenes (inability to produce functional mRNA and proteins) [133]. In addition, GJA1 is ubiquitously expressed in many human tissues and cells [134]. 

Comment: The authours mention the chromosome location of many, but not all, of the chromosome locations of the connexin genes. For example, it is mentioned in the manuscript that Cx30 is found on chromosome 13 but it is not mentioned where Cx26 is located. It would be useful to mention that Cx30 and Cx26 are close together which would strengthen the argument for Cx30 not being involved in hearing impairment as much.

Response: The chromosomal position of the Cx genes are now added to the text, as follows:

 Line 282: “Connexin 26 gene (GJB2, OMIM:121011) located on chromosome 13q12.11 is known to be expressed in different tissues including the cochlear of human [38], mouse and rat [39].”

The text has also been updated to include the fact that Cx 30 and Cx 26 are located at the same position on chromosome 13:

Line 386: “The location of GJB6 in the human genome is chromosome 13q12.11 (same as GJB2) …”

Comment: With some connexins the authours mention the localization and function of the subtypes in hearing but other are left out. I would recommend that the localization in the auditory tract for all seven connexins examined in the study be defined in the study. Since this study is examining human patients, it would be advisable to also include the human localization studies of which many of these connexin subtypes have been examined in terms of localization in the ear.

Response: the localization of the connexins has been added to the discussion as follows:

As stated above:

Line 282: Connexin 26 gene (GJB2, OMIM:121011) located on chromosome 13q12.11 is known to be expressed in different tissues including the cochlear of human [58], mouse and rat [59].

Line 386: The location of GJB6 in the human genome is chromosome 13q12.11 (same as GJB2). The GJB6 gene is expressed mainly in the brain and skin with about 76% protein identity when compared to human GJB2.

Line 417: In addition, GJA1 is ubiquitously expressed in many human tissues and cells [134].

Comment: Finally, it would be valuable to have a final figure in the main manuscript summarizing whether each connexin is important in hearing or not. For some subtypes the authours mention that there are confounding reports. However, it is not clear to me what the final conclusions are after performing this comprehensive analysis. A final figure/table/schematic outlining each connexin subtype and their relation to hearing impairment would complement the findings of the study.

Response: Thanks for the excellent suggestion: The paragraphs below has been inserted at the end of the discussion, and a sentence in the abstract and conclusion the manuscript to clearly define GJB2 and GJB3 as the only recognized and established HI-associated genes.

Line 471:

There was a limited numbers publications retrieved on GJB3, GJB4, GJC3, GJA1, and GJC1 which was a major challenge encountered in this study. It was therefore difficult to determine the whether these genes should be considered or catalogued as HI gene. Apart from GJB2, which is globally known to be associated HI, there is a need for more studies from different populations as well as functional studies for a concluding decision to be made on GJB4, GJA1, GJC3, and GJC1. The Hearing Loss Home Page [8] has categorized GJB3 as an autosomal dominant HI gene. Although GJB6 was previously considered as HI gene, recent studies have disproved its classification as a HI gene. The present review suggests that only GJB2 and GJB3 are recognized and validated HI genes.

.

Minor:

1) Line 51-53: This is a confusing sentence, consider rewording

Response: The sentence now reads:

Line 53:  It is estimated that approximately 50% of congenital profound HI cases are of genetic origin [5].

2) Line 53: Change “distinguish” to “distinguishing”

Response:

Line 55: “distinguish” changed to “distinguishing

3) Line 91: [1] is in a different reference format then the rest of the introduction

Response: The reference format has been corrected

4) Line 108: Change “extract” to “extracted”

Response:

Line 104: Extract changed to “extract” to “extracted”

5) Figure 1C: The number of records is small and hard to see. The different shades of blue are also difficult to differentiate

Response: The figure has been replaced with an updated version that shows the differences clearly.

6) Line 118: Change “compare” to “compared”

Response:

Line 114: “compare” changed to “compared”

7) Line 119: Change “continent” to “continents”

Response:

Line 115: “continent” changed to “continents”

8) Line 116: In figure 4 legend. Should read “global distribution of common…”

Response:

Line 166: The legend has been changed accordingly

9) All tables are cut off in PDF so cannot see the first and last of it. Consider addressing.

Response: The tables are now resized to show their content in full

10) Line 219: Change “was” to “were”

Response: “was” changed to “were”

11) In the discussion the style of references has changed. This needs to be consistent throughout. 

Response: The referencing style are updated through the manuscript.

12) Line 276: Change “migrations” to “migration”

Response:

Line 276: “migrations” changed to “migration”

13) Line 327: Change to Introduced

Response:

Line 358: “introduce” changed to “introduced”

14) Line 334: There is a “.” After gene. Get rid of the word “impaired”

Response: the word “impaired” is deleted

15) Line 338: Should read “and hence is responsible…”

Response: The line is revised accordingly

16) Line 339: Change “mice” to “mouse”

Response:

Line 401: changed, accordingly.

17) Line 339: Should mention here that the original Cx30 knock-out mouse also reduced Cx26 expression and that was why they had hearing loss. Tuebner et al. 2003, Connexin 30 deficiency causes severe impairment and lack of endocochlear potential, Genetics should be cited.

Response: The discussion has been updated with the line below:

Line 400:  The previously reported GJB6 knockout mice had significant reduction in the GJB2 expression which was the cause of HI [125].

18) Line 354: Change toIt is possible…”

Response:

Line 422:  The sentence is changed accordingly

19) Line 356: Need a reference here, what evidence is there for this theory?

Response: Reference provided

20) Line 358: “Encodes” is misplaced, revise.

Response: the word encodes is deleted

21) Line 403: “also” is written twice, revise.

Response: Revision made accordingly

22) Line 440: Change “unto” to “onto”.

Response: “unto” changed to “onto"

Reviewer 2 Report

Remarks :

Page 2 line 45-50 there is need to put some more articles about that epidemiological data, more citations with more data .

There is need to explain more differences and similarities between different groups as for non genetic specialists

Author Response

Reviewer 2

Page 2 line 45-50 there is need to put some more articles about that epidemiological data, more citations with more data.

Response: The text has been updated accordingly with epidemiological data.

Line 48:

It occurs in about 1 per 1000 live births in high income countries, with a much higher incidence of up to 6 per 1000 in the lower income countries [3]. According to the World Health Organization, 466 million people are living with HI and about 900 people will be affected by the year 2050 [4]. Depending on the degree of severity, HI can be classified as mild, moderate, severe, or profound when the pure tone average ranges from 26 to 40 dB, 41 to 60 dB, 61 to 80 dB or is over 81 dB, respectively (Deafness and Hearing Loss, n.d.). It is estimated that approximately 50% of congenital profound HI cases are of genetic origin [5].

There is need to explain more differences and similarities between different groups as for non-genetic specialists

Response: The discussion has been improved accordingly especially for Cx26 and Cx30

Reviewer 3 Report

Comments are available in attached file

Author Response

Reviewer 3

In an article Samuel M. Adadey and co-authors «Connexin Genes Variants Associated with

Non-Syndromic Hearing Impairment: A Systematic Review of the Global Burden»

The article by Samuel M. Adadey and co-authors is regard to the conducted a

systematic review of the literature based on targeted inclusion/exclusion criteria of publications from 1997 to 2020. The databases used were PubMed, Scopus, Africa-Wide Information and Web of Science. So, for this study, 571 independent scientific articles were extracted from the presented databases into a systematic review, in which the results of studies of connexin genes associated with hearing impairment/loss in various countries of the world were published.

The topic of the article is very relevant and interesting to the scientific community on the problem of hereditary hearing impairments and has prospects for further research.

Response: thanks for the reviewer’s positive comments.

Major comment

  1. For a comparative analysis of the results obtained in the study, the authors did not discuss similar works. We consider it a mistake to include such studies in the exclusion criteria, since they are a model example for a review or meta-analysis of studies related to hereditary hearing impairments. Currently, these are two works:

1) Chan, D. K., & Chang, K. W. (2013). GJB2-associated hearing loss: Systematic review of worldwide prevalence, genotype, and auditory phenotype. The Laryngoscope, 124(2), E34–E53.

doi:10.1002/lary.24332

2) Tsukada, K., Nishio, S., Hattori, M., & Usami, S. (2015). Ethnic-Specific Spectrum of GJB2 and

SLC26A4 Mutations. Annals of Otology, Rhinology & Laryngology, 124(1_suppl), 61S–76S.

doi:10.1177/0003489415575060

Thus, in 2013, the data of a meta-analysis of 216 scientific articles were published containing the results of molecular genetic studies of the GJB2 gene in hearing impaired probands from different countries of the world (Chan et al., 2013). As a result of this metaanalysis, a total of 43530 probands from 63 countries were covered, where 7518 individuals were found to have biallelic pathogenic variants of the GJB2 gene associated with hearing loss. Thus, the contribution of pathogenic variants of the GJB2 gene to the etiology of hearing loss in the world, on average, was 17.3%. Of the countries presented, the highest contribution of pathogenic variants was shown in Europe (27.1%), and the lowest contribution was in sub-Saharan Africa (5.6%) (Chan et al., 2013).

In 2015 the results of a meta-analysis on the spectrum and frequency of the identified pathogenic variants of the GJB2 gene and the SLC26A4 gene in different populations of the world from the point of view of ancient human migrations were announced. Subsequently, after a cluster analysis of these two genes, similar population scenarios were revealed in the distribution of major mutations in the GJB2 gene and the SLC26A4 gene (Tsukada et al., 2015).

Response: Like the two reviews listed in the reviewer’s comment above, review articles usually would include only original articles hence, we excluded the two reviews mentioned above during the literature search and data extraction. The two studies however have been considered in the discussion, as follows:

Line 290:

In a previous review by Chan and Chang in 2014, 216 original GJB2 research articles reporting not less than 10 proband were retrieved and analyzed [63]. In our current review, 571 original research publications on connexins associated with HI were considered of which 566 articles reported on GJB2 associated HI. The previous report was from 63 countries [63], while in this study, we retrieved GJB2 publications from 106 countries. The differences in the number of publications and countries involved can be explained by the time difference between the previous report and the present study. In addition, we did not exclude case reports, contrary to the previous report. In contrast to the report from Chan and Chang, Australia, and not Africa, had the lowest contribution of GJB2 variants. Moreover, Asia was identified as the highest contributor while the previous report had Europe as the highest contributor of GJB2 PLP to HI [63]; this can be attributed to the increasing  interest and numbers of Genetic researches in all part of the world. Despite the above differences, the commonly reported PLP GJB2 variants were similar in both studies. Furthermore, our study and the studies from Chan and Chang and Tsukada reported similar ethic specific spectrum of the common PLP variants in GJB2 [63,64].  

  1. The study was carried out incorrectly, since the results do not correspond to some of the goals or objectives of the study. The material presented does not correspond to the following text:

…. Using a systematic review approach, we provided a summary data on the

distribution of connexin gene variants, and their contribution to NSHI in various

populations around the world….

And at the beginning of the discussion:

…. To our knowledge, this paper is, to date, the most comprehensive review on the contribution of connexin genes variants in HI, globally….

In this work, the contribution of common pathogenic variants to the etiology of hearing loss separately by continent/country and in the world as a whole is not assessed, no calculations have been made for this. It is known that it is on the basis of assessing the contribution of mutations (i.e., counting homozygous and compound-heterozygous carriers) that one can find out the spread of hereditary deafness in the world/countries due to certain mutations of any genes. The authors have declared an unfulfilled task.

We consider it necessary, on the basis of factual material (the literature under review), to carry out calculations on the contribution and frequency of the most common mutations clearly across continents, with the identification of countries and/or populations with the highest frequency of distribution in them, followed by identification of the frequency of the variant in the world as a whole. This task is fundamental for such meta-analytical studies concerning the problems of genetically determined diseases.

We propose to present the results of the study in the form of a table, an example of which is given below:

Response:

The aim of the study was rephrased to reflect the results, as follows:

Line 91:

Using a systematic review approach, we provided a summary data on connexin gene variants associated with HI, and specifically the global contribution of connexin genes to NSHI.

Thank you for suggesting the table as a guide.

Line 255:

We have provided a table (Table 7) to describe the contribution of the common PLP connexin variants to HI and we used the data to update the manuscript, as appropriate.

We calculated the allele frequency of the reported PLP variants in connexin genes in patients and controls for each continent, in order to assess the global contribution of these variants associated to HI. Asia had the highest allele frequencies of the common GJB2 variants in the hearing controls compared to the other continents. GJB2:p.V37I:c.109G>A had the highest allele frequency (200/3478 (5.8%)) among the control group from Asia (200/3478 (5.8%)) and North America (11/588 (1.9%)). In Africa, GJB2:p.Gly12ValfsTer2:c.35delG and p.R143W:c.427C>T recorded the highest frequencies in Ghana. In Europe, among the eight common GJB2 variants, p.Gly12ValfsTer2:c.35delG (Table 7). We did not find reports of carriers of the GJB6 large deletions and GJA1 PLP variants.   

  1. In an overview of the GJB2 gene variants:

1) the frequency of 8 common options is not calculated as a percentage, but is shown

quantitatively, this makes it difficult to perceive the results obtained:

….p.Gly12ValfsTer2 (c.35delG ) was ranked as the most commonly reported GJB2

variant with 26429 mutated alleles out of 175491 investigated alleles. The frequency of the other common GJB2 mutations were 10009/82805, 152 9813/277116, 3520/127802, 1080/107855, 972/54394, 641/54279, and 497/90305 for p.M34T (c.101T>C), 153 p.L79Cfs (c.235delC), p.V37I (c.109G>A), p.H100RfsTer14 (c.299_300delAT), p.W24* (c.71G>A), p.L56Rfs (c.167delT), and p.R143W (c.427C>T) respectively (Table S1)…;

Response:

The frequencies have been calculated as percentages and it reads:

Line 147:

 Based on the number of reported alleles, p.Gly12ValfsTer2 (c.35delG ) was ranked as the most commonly reported GJB2 variant,  found in  26429 (15.1%) out of 175491 investigated alleles. The frequency of the other common GJB2 mutations were 10009/82805 (12.1%), 9813/277116 (3.5%), 3520/127802 (2.8%), 972/54394 (1.8%), 641/54279 (1.2%), 1080/107855 (1.0%), and 497/90305 (0.6%) for p.M34T (c.101T>C), p.L79Cfs (c.235delC), p.V37I (c.109G>A), p.W24* (c.71G>A), p.L56Rfs (c.167delT), p.H100RfsTer14 (c.299_300delAT), and p.R143W (c.427C>T), respectively (Table S3).

2) among the spectrum of pathogenic gene variants, the authors did not consider the known rare GJB2 mutations that are characteristic only for certain (isolated) ethnic groups, which is a big omission in the context of the problem under consideration. For example, among the Ashkenazi Jews (Israel), there is a deletion c.167delC (p.Leu56Argfs), among the Maya Indians in Central America (Guatemala), the nonsense substitution prevails - c.131G>A (p.Trp44Ter), in South America there are: c. 19C>T (p.Gln7Ter) (Ecuador) and s.596C>T

(p.Ser199Phe) (Colombia): Brownstein, Z. Deafness genes in Israel: implications for diagnostics in the clinic / Z. Brownstein, K.B. Avraham // Pediatr Res. - 2009. - Vol. 66(2). - Р. 128-34. doi: 10.1203/PDR.0b013e3181aabd7f; Tamayo, M.L. Molecular studies in the GJB2 gene (Cx26) among a deaf population from Bogota, Colombia: results of a screening program / M.L. Tamayo, M. Olarte, N. Gelvez, M. Gómez, J.L. Frías, J.E. Bernal, S. Florez, D. Medina // Int J Pediatr Otorhinolaryngol. - 2009. - Vol. 73(1). - Р. 97-101Paz-y-Miño, C. Frequency of GJB2 and del(GJB6-D13S1830) mutations among an Ecuadorian mestizo population / M.B. Petersen C. Paz-y-Miño. D. Beaty, A. López-Cortés, I. Proaño // Int J Pediatr Otorhinolaryngol. 2014 - Vol. 78(10). - Р. 1648-1654. doi: 10.1016/j.ijporl.2014.07.014; Carranza, C. Mayan founder mutation is a common cause of deafness in Guatemala / C. Carranza, I. Menendez, M. Herrera, P. Castellanos, C. Amado, F. Maldonado, L. Rosales, N. Escobar, M. Guerra, D. Alvarez, J. II Foster, S. Guo, S.H. Blanton, G. Bademci, M. A Tekin // Clin Genet.- 2016. - Vol. 89. - Р. 461-465. doi: 10.1111/cge.12676. In Russia, among Siberian populations, unique allelic variants of the GJB2 gene were found. One of them is c.-23+1G>A (IVS1+1G>A) found in Yakut patients (Eastern Siberia) with a frequency of 16.2 people per 100,000 indigenous populations, and the frequency of heterozygous carriage c.-23+1G>A it was 3% -11%. Thus, the largest focus of the world accumulation of the pathogenic GJB2 variant, c.-23+1G>A, was recorded, due to the founder's

effect (Barashkov et al, 2011): Barashkov, N.A. Age-Related Hearing Impairment (ARHI) Associated with GJB2 Single Mutation IVS1+1G>A in the Yakut Population Isolate in Eastern Siberia / N.A. Barashkov, F.M. Teryutin, V.G. Pshennikova, A.V. Solovyev, L.A. Klarov, N.A. Solovyeva, A.A. Kozhevnikov, L.M. Vasilyeva, E.E. Fedotova, M.V. Pak, S.N. Lekhanova, E.V. Zakharova, K.E. Savvinova, N.N. Gotovtsev, A.M. Rafailo N.V. Luginov, A.N. Alexeev, O.L. Posukh, L.U. Dzhemileva, E.K. Khusnutdinova, S.A. Fedorova // PLoS One. - 2014. - Vol. 9(6). - e100848;

Barashkov, N.A. Spectrum and Frequency of the GJB2 Gene Pathogenic Variants in a Large Cohort of Patients with Hearing Impairment Living in a Subarctic Region of Russia (the Sakha Republic) / N.A. Barashkov, V.G. Pshennikova, O.L. Posukh, F.M. Teryutin, A.V. Solovyev, L.A. Klarov, G.P. Romanov, N.N. Gotovtsev, A.A. Kozhevnikov, E.V. Kirillina, O.G. Sidorova, L.M. Vasilyevа, E.E. Fedotova, I.V. Morozov, A.A. Bondar, N.A. Solovyevа, S.K. Kononova, A.M. Rafailov, N.N. Sazonov, A.N. Alekseev, M.I Tomsky, L.U. Dzhemileva, E.K. Khusnutdinova, S.A Fedorova // PLoS One. - 2016. - Vol.11(5):e0156300. doi: 10.1371/journal.pone.0156300.

A striking discovery was the high prevalence of the rare variant c.516G> C

(p.Trp172Cys) in Tuvinians (Southern Siberia), accounting for 62.9% of all mutant GJB2 alleles and a carrier frequency of 3.8% in controls (Posukh et al, 2019): Olga L Posukh, Marina V Zytsar, Marita S Bady-Khoo, Valeria Yu Danilchenko, Ekaterina A Maslova,

Nikolay A Barashkov, Alexander A Bondar, Igor V Morozov, Vladimir N Maximov, Michael I Voevoda. Unique Mutational Spectrum of the GJB2 Gene and its Pathogenic Contribution to Deafness in Tuvinians (Southern Siberia, Russia): A High Prevalence of Rare Variant c.516G>C (p.Trp172Cys). 2019 Jun 5;10(6):429. doi: 10.3390/genes10060429. PMID: 31195736 PMCID: PMC6627114 DOI: 10.3390/genes10060429.

The carrier frequencies of с.35delG and c.101T>C were identical (2.5%) in the Russian control group. We found that the contribution of the GJB2 gene pathogenic variants in HI in the population of the Sakha Republic (48.85%) was the highest among all of the previously studied regions of Asia. We suggest that extensive accumulation of the c.-23+1G>A pathogenic variant in the indigenous Yakut population (92.20% of all mutant chromosomes in patients) and an extremely high (10.20%) carrier frequency in the control group may indicate a possible selective advantage for the c.-23+1G>A carriers living in subarctic climate. (Barashkov et al., 2016)

Response: Thanks for the insightful comments. We have reanalyzed our meta data to include the isolated populations as well as the variants common to them. This has been discussed thoroughly in the text and tabulated in the supplementary documents.

Line 140:

Asia (n=288, 48%) was the highest contributor of PLP followed by Europe (n= 166, 28%), North America (n=52, 9%), Africa (n=48, 8%), South America (n=37, 6%) and Australia (n=4, 1%).

Line 149:

Based on the number of reported alleles, p.Gly12ValfsTer2 (c.35delG ) was ranked as the most commonly reported GJB2 variant with 26429 (15.1%) mutated alleles out of 175491 investigated alleles. The frequency of the other common GJB2 mutations were 10009/82805 (12.1%), 9813/277116 (3.5%), 3520/127802 (2.8%), 972/54394 (1.8%), 641/54279 (1.2%), 1080/107855 (1.0%), and 497/90305 (0.6%) for p.M34T (c.101T>C), p.L79Cfs (c.235delC), p.V37I (c.109G>A), p.W24* (c.71G>A), p.L56Rfs (c.167delT), p.H100RfsTer14 (c.299_300delAT), and p.R143W (c.427C>T) respectively (Table S3).

Line 171:

We extracted data on known PLP GJB2 varaints (p.W44*: c.131G>A (North America), c.IVS1+1G>A (Russia), p.W172*: c.516G>A (Siberia), p.W172C: c.516G>C (Siberia)  p.W172R , and c.514T>A (Siberia) that are prevalent in isolated ethinic groups. The majority of mutated c.IVS1+1G>A were recorded in Asia with high frequencies from Yakutia, Seria and Rusia  (Figure S3). The other 3 rare GJB2 variants (p.W172*: c.516G>A, p.W172C: c.516G>C, p.W172R , and c.514T>A) which are at the same amino acid position were common in the Asian countries with Rusia having the highest frequency (Table S4).    

Line 369:

Rare known PLP GJB2 variant were found to displace some degree of ethnic specificity in some populations. Similar to previous reports [101,102], we identified extremely high frequencies of GJB2: c.IVS1+1G>A from Yakutia and Russia. The high prevalence of the variant within the Yakutia population can be seen in the high carrier of 10.02 [102]. This is also suggestive of a potential accumulation of GJB2 pathogenic variant in this population and calls for public health attention [101]. The Siberian population which comprises of Russia, Kazakhstan boarders of Mongolia and China was found to have the rare GJB2: p.W172C (c.516G>C). We analyzed other variant (p.W172* (c.516G>A), p.W172R  (c.514T>A)) at the same site of mutation, and similar to our observation, a previous study reported extremely high frequency of 62.9% from these populations [103]. Other population specific GJB2 rare variants found were p.W44*, p.Q7*, and p.S199F which are common to the North America [75,95,104], Ecuador [105] and Colombia [106] respectively.

3) Despite the fact that the GJB2 gene is the most studied, the authors have not

discussed the works concerning the search for mutations in the intron region adjacent to exon 1 of the GJB2 gene. In this area, for example, there is a recessive mutation of the splicing site c.-23+1G>A (IVS1+1G>A), which is often found in Turkish speaking populations (Mongolia, Turkey, Russia and many other countries). It is interesting how often research is carried out in the world to search for mutations in this area, why there are few such studies.  

Response: Thank you for the comment. We have introduced a paragraph to discuss pathogenic intronic variant within the GJB2 non-coding region.

Line 380:

Although not in high numbers, we identified pathogenic GJB2 intronic variants that were reported in hearing impaired patients. These variants were c.IVS1 -1G>A [107], c.IVS1-15C>T [75], c.IVS1+12G>A [108], c.IVS1+27G>C [109], and c.-23+1G>A. The -23+1G>A variant was reported in studies from Russia [110], Poland [111], China [112], India [91], United States of America [113], and with a high prevalence from Turkey [114].      

  1. In the GJB6 gene, about 20 mutations are known to date. But in this work, only one common deletion del(GJB6-D13S1830) was more or less affected. Currently, 6 large GJB6-deletions have been announced in the HGMD database (http://www.hgmd.cf.ac.uk/ac/all.php).

The authors ignored new interesting works concerning deletions affecting connexin genes, which were not included in the list of references:

1) Bliznetz, E.A., Lalayants, M.R., Markova, T.G., Balanovsky, O.P., Balanovska, E.V., Skhalyakho,

R.A. . . . Polyakov, AV. (2017). Update of the GJB2/DFNB1 mutation spectrum in Russia: a founder

Ingush mutation del(GJB2-D13S175) is the most frequent among other large deletions. J Hum Genet.

62(8):789-795. doi: 10.1038/jhg.2017.42.

Figure from the article: Bliznetz et al, 2017.

Аnd article by Arti Pandya and co-authors is regard to the analysis of the mutation spectrum of the GJB2 gene and the analysis of two common deletions of DFNB1 loci -del(GJB6-D13S1830) and del(GJB6-D13S1854) in a large cohort of ethnically diverse deaf probands (n = 2376) from the USA:

2) Arti Pandya, Alexander O'Brien, Michael Kovasala, Guney Bademci, Mustafa Tekin, Kathleen S

Arnos. Analyses of del(GJB6-D13S1830) and del(GJB6-D13S1834) deletions in a large cohort with

hearing loss: Caveats to interpretation of molecular test results in multiplex families. Mol Genet Genomic Med. 2020 Apr;8(4):e1171. doi: 10.1002/mgg3.1171. Epub 2020 Feb 17. Affiliations expand. PMID: 32067424 PMCID: PMC7196463 DOI: 10.1002/mgg3.1171

Response: Thank you for prompting us. We have included all the six deletions and an additional (seventh) one in the discussion. The publications Pandya et al and Bliznetz have been included in text.

Line 391:

Six GJB6 large genomic deletions have been found and previously reported, they are: >920kb deletion [118], 179 kb deletion [119], 131 kb deletion [120], del(GJB2-D13S175), del(GJB6-D13S1830) and del(GJB6-D13S1854) [121]. In addition to the six, there was a report of a seventh GJB6 deletion, del(GJB6‐D13S1834), in February 2020 [122].

  1. A big drawback of the work is the controversial point regarding variants with not proven clinical significance of pathogenity (Gene variants GJB3, GJB4, GJC3, GJA1, GJC1). In the context of the problem, these options should not be considered at all, or should have had a detailed discussion. Thus, the authors mislead the reader. As stated by the authors, only pathogenic/likely pathogenic variants of connexin genes associated with hearing impairment should be considered in the inclusion criteria.

Response: The comment above has been considered as a limitation and discussed accordingly in the manuscript, and amend, the abstract and conclusion, accordingly.

Line 370:

There was a limited numbers publications retrieved on GJB3, GJB4, GJC3, GJA1, and GJC1 which was a major challenge encountered in this study. It was therefore difficult to determine the whether these genes should be considered or catalogued as HI gene. Apart from GJB2, which is globally known to be associated HI, there is a need for more studies from different populations as well as functional studies for a concluding decision to be made on GJB4, GJA1, GJC3, and GJC1. The Hearing Loss Home Page [8] has categorized GJB3 as an autosomal dominant HI gene. Although GJB6 was previously considered as HI gene, recent studies have disproved its classification as a HI gene. The present review suggests that only GJB2 and GJB3 are recognized and validated HI genes.

  1. In the work, the authors cite 7 times the work of Pfenniger et al., 2011, but this article is absent in two references.

Response: The citation and reference list has been updated accordingly (ref. 11)

Reviewer 4 Report

This is a comprehensive review on the contribution of rare variation in connexin genes to hearing loss. The methodology is very well explained and the results are clearly presented. The major value is the geographical distribution of connexin variants across different populations worldwide.

I would suggest including a short section in the discussion about rare variants in GJB2 and Meniere disease (MD), a condition defined by sensorineural hearing loss, tinnitus and episodic vertigo. A burden of rare missense variants has been reported in the coding regions of GJB2 in a large cohort of patients with sporadic MD, and these variants may contribute to hearing loss in MD (Gallego-Martinez et al, 2019). Moreover, the low prevalence of MD in sub-saharan population could be related with the differences in the allelic frequencies of rare variants in hearing loss genes, including GJB2.

Some minor editing is needed. Some abbreviations are not used systematically in the text like PLP (pathogenic, likely pathogenic) or PCR that are detailed several times.

Author Response

Reviewer 4

This is a comprehensive review on the contribution of rare variation in connexin genes to hearing loss. The methodology is very well explained, and the results are clearly presented. The major value is the geographical distribution of connexin variants across different populations worldwide.

Response: thanks very much for your positive comments.

I would suggest including a short section in the discussion about rare variants in GJB2 and Meniere disease (MD), a condition defined by sensorineural hearing loss, tinnitus and episodic vertigo. A burden of rare missense variants has been reported in the coding regions of GJB2 in a large cohort of patients with sporadic MD, and these variants may contribute to hearing loss in MD (Gallego-Martinez et al, 2019). Moreover, the low prevalence of MD in sub-saharan population could be related with the differences in the allelic frequencies of rare variants in hearing loss genes, including GJB2.

Response: The discussion has been revised as follows

Line 361:

In our current review, we identified some rare GJB2 variants that were absent from the databases used. Although these variants were identified in hearing-impaired participants, they may not be considered as HI variants since there is no functional and/or population studies on these variants. A recent report showed that rare within the coding region of GJB2 gene and other HI genes are associated with Meniere disease (MD) [98]. The signs and symptoms of MD are sensorineural hearing loss, tinnitus and episodic vertigo which are sometimes common to patients with NSHI [99]. Meniere disease was reported to be extremely rare in individuals from African decent which supported the negligible contribution of GJB2 variants to HI in sub-Saharan Africa [100].  

Some minor editing is needed. Some abbreviations are not used systematically in the text like PLP (pathogenic, likely pathogenic) or PCR that are detailed several times.

Response: The text has been edited to systematically use the abbreviations.

Round 2

Reviewer 3 Report

In the text there are many grammatical errors, this must be carefully checked